# Enhanced jet stream waviness induced by suppressed tropical Pacific convection during boreal summer

Xiaoting Sun [1,2,3,4], Qinghua Ding [3✉], Shih-Yu Simon Wang [5], Dániel Topál[3,6,7], Qingquan Li[1,4], Christopher Castro[8], Haiyan Teng [9], Rui Luo[10,11] & Yihui Ding[4]

Consensus on the cause of recent midlatitude circulation changes toward a wavier manner in the Northern Hemisphere has not been reached, albeit a number of studies collectively suggest that this phenomenon is driven by global warming and associated Arctic amplification. Here, through a fingerprint analysis of various global simulations and a tropical heating-imposed experiment, we suggest that the suppression of tropical convection along the Inter Tropical Convergence Zone induced by sea surface temperature (SST) cooling trends over the tropical Eastern Pacific contributed to the increased summertime midlatitude waviness in the past 40 years through the generation of a Rossby-wave-train propagating within the jet waveguide and the reduced north-south temperature gradient. This perspective indicates less of an influence from the Arctic amplification on the observed mid-latitude wave amplification than what was previously estimated. This study also emphasizes the need to better predict the tropical Pacific SST variability in order to project the summer jet waviness and consequent weather extremes.

[1] Key Laboratory of Meteorological Disaster, Ministry of Education/Joint International Research Laboratory of Climate and Environment Change/ Collaborative Innovation Center on Forecast and Evaluation of Meteorological Disasters, Nanjing University of Information Science and Technology, 210044 Nanjing, China. [2] Chinese Academy of Meteorological Sciences, 100081 Beijing, China. [3] Department of Geography and Earth Research Institute, University of California, Santa Barbara, CA, USA. [4] Laboratory for Climate Studies, National Climate Center, China Meteorological Administration, Beijing, China. [5] Department of Plants, Soils, and Climate, Utah State University, Logan, UT, USA. [6] Institute for Geological and Geochemical Research, Research Centre for Astronomy and Earth Sciences, Eötvös Loránd Research Network, Budapest, Hungary. [7] ELTE Eötvös Loránd University, Doctoral School of Environmental Sciences, Budapest, Hungary. [8] Department of Hydrology Atmospheric Sciences, The University of Arizona, Tucson, AZ, USA. [9] Pacific Northwest National Laboratory, Richland, WA, USA. [10] Deep-Sea Multidisciplinary Research Center, Pilot National Laboratory of Marine Science and Technology (Qingdao), 266237 Qingdao, China. [11] Frontiers Science Center for Deep Ocean Multi-spheres and Earth System, Key Laboratory of Physical Oceanography, Ocean University of China, 266100 Qingdao, China. ✉email: Qinghua@ucsb.edu

Atmospheric scientists have expressed concern about the increasing meandering of extratropical atmospheric circulations during the boreal summer (referring to June-July-August, JJA), which has accompanied extreme weather events in recent decades[1–6]. Prevalent heatwaves and flood-producing storms across the midlatitudes have been linked to enhanced amplitudes of atmospheric quasi-stationary wave trains along the jet stream, especially over North America and Eurasia[7–12]. These amplified quasi-stationary waves exhibit a conspicuous circumglobal feature with a predominant zonal wavenumber 5–7 structure[4,13] and are particularly resonant with the boreal summer Rossby-wave propagation mode on inter-annual and interdecadal timescales[14–16]. However, conflicting hypotheses that have emerged to explain the amplified circumglobal wave train has made the attribution of this anomalous circulation feature inconclusive and its projection difficult[17–21].

Two schools of thoughts dominate the prevailing theories explaining the amplified circumglobal wave train: one idea attributes the amplification of circumglobal short waves to the anthropogenically induced changes in the high-latitude climate system, such as Arctic amplification (AA)[22]. Anchored in the theory that the meridional temperature gradient controls the waviness of the jet[5,23–25], AA increases midlatitude stationary waves via the reduction of the equator-to-pole thermal gradient, which acts to shift the jet stream while enhancing its sensitivity to extratropical diabatic heating anomalies[26–29]. This hypothesis has undergone rigorous scientific scrutiny, since some studies showed contrasting results and speculated that the increased waviness is the cause of the meridional temperature gradient changes and not the other way around[30]. The other theory concerns the role of the change in diabatic heating associated with sea surface temperature (SST), soil temperature, and anomalous convection in the tropics and extratropics in perturbing quasi-stationary wave trains along the jet stream and subsequent waviness[31–36]. The scientific community has not reached a consensus on the aforementioned two sources of increased jet stream waviness, nor has it acquired a complete understanding of the sensitivity of various properties (e.g., wavelength, amplitude, and vertical structure) of midlatitude stationary waves to different climate drivers. The purpose of this research is to investigate these two perspectives and shed light on the profound midlatitude circulation changes alongside the concurrent Arctic warming and anthropogenic forcing.

## Results

### Observed and simulated midlatitude summer circulation trends.
We first describe the observed global summer atmospheric circulation changes over the satellite era (1979–2018) and compare them to state-of-the-art climate model simulations. The Northern Hemispheric (NH) summer atmospheric circulation has exhibited prominent changes over the past 40 years, as exemplified by the linear trend pattern of the 200-hPa geopotential height (Z200) in Fig. 1a. A chain of isolated height centers along the westerly jet stream (~45°N) is visible. With the zonal mean component removed (hereafter "non-zonal", see Methods section), a clear circumglobal wave train manifests as successive high- and low-pressure trending anomalies around 45°N (Fig. 1b). This wave train shows a pronounced zonal wavenumber-5 structure echoing a recurrent mode of summertime atmospheric circulation, the circumglobal teleconnection (CGT) on interannual and interdecadal timescales[15]. Notably, the amplitude of the zonal-mean Z200 trend in the NH midlatitudes is approximately double of that in the tropics (Fig. 1d). To better describe the increasing waviness, we show the trend of meridional winds (V200) in Fig. 1c, which also depicts an enhancement along

the jet waveguide that is consistent with the non-zonal Z200 trend pattern.

Concomitant changes in global SSTs exhibit widespread warming almost everywhere in the NH, with the strongest trend surrounding the Arctic and the NH extratropics (Fig. 1e). However, little-to-no change in SSTs is observed in the tropical Eastern Pacific (TEP), while the Pacific sector of the Southern Ocean shows significant cooling (Fig. 1e). To better reveal internal variability of SSTs masked by the global warming signal, we remove the global mean SST (averaged between 60°S to 60°N) from Fig. 1e (hereafter "non-global SST") to highlight the fact that the TEP has experienced widespread relative cooling over the satellite era (Fig. 1f). This cooling pattern is reminiscent of the negative phase of interdecadal SST modes of variability in the Pacific, such as the Pacific Decadal Oscillation (PDO)[37] or the Interdecadal Pacific Oscillation (IPO)[38]. The accompanying long-term trend of JJA precipitation in the tropics reveals a narrow and elongated drying zone along the Inter Tropical Convergence Zone (ITCZ) (~5°N) extended from the Indochina Peninsula to the eastern Pacific[39,40]. Meanwhile, the Maritime continent is the only area showing increased rainfall in the deep tropics (Fig. 1g).

By contrast, the observed changes in atmospheric circulation and SST patterns are not seen in our best estimate of climate response to anthropogenic forcing based upon the ensemble mean of the CESM1 Large Ensemble (CESM-LE) and multiple climate models from CMIP5 and CMIP6. The linear trend of the CESM-LE mean Z200 in the historical experiment (1979–2018) shows rather uniform height increases all over the NH, especially in the tropics (Fig. 2a), with only feeble waviness along the NH jet as indicated by the non-zonal Z200 (Fig. 2b) and V200 linear trends (Fig. 2c). Both CMIP5 (Supplementary Fig. 1) and CMIP6 models (Supplementary Fig. 2) also reveal little waviness in their respective multi-model means. The amplitude ranges of these simulated non-zonal Z200 and V200 trends are between $-4$–4 m decade$^{-1}$ and $-0.3$–0.3 m s$^{-1}$ decade$^{-1}$, respectively, which are about one-third of the observed ranges ($\pm12$ m decade$^{-1}$ and $\pm1$ m s$^{-1}$ decade$^{-1}$) in the NH midlatitudes. Remarkably, all members of the CESM-LE and CMIP6, and most of the CMIP5 models, overestimate the zonal mean component of Z200 trend between 30°S and 60°N (Fig. 2d and Supplementary Figs. 1d and 2d), with especially large positive bias in the tropics. This is indicative of overestimated tropical warming trends in current climate models under observed anthropogenic forcing[41]. Accordingly, the SST trend patterns in the CESM-LE and CMIP5/6 multi-model ensemble means reflect prominent global ocean warming, especially in the Arctic, suggesting that the models capture the enhanced NH high-latitude warming associated with AA while failing to faithfully represent change in the tropics (Supplementary Figs. 1f and 2f).

There is a marked warming trend in the tropical Eastern Pacific (TEP) in the ensemble mean simulations (0.02 K/decade, Fig. 2f) suggesting that anthropogenic forcing favors El-Nino-like warming in the tropical Pacific, rather than the observed TEP cooling[42,43] ($-0.06$ K/decade; Fig. 1f vs. 2f). In parallel to the simulated tropical SSTs, both the CESM-LE mean (Fig. 2g) and the multi-model mean of CMIP5/6 simulations (Supplementary Figs. 1g and 2g) struggle to replicate the observed rainfall trend pattern. The strengthened ITCZ simulated by these climate models in responses to anthropogenic forcing (Fig. 2g) is a stark contrast to what is seen in the observations[44] (Fig. 1g).

The discrepancy between the SST and precipitation trends over the past 40 years in the observed and simulated global circulations leads to our hypothesis that anthropogenic forcing and associated AA alone may not suffice to explain the summertime midlatitude wave amplification in the observations. The ensuing analysis therefore explores how reduced convection in the tropical West-

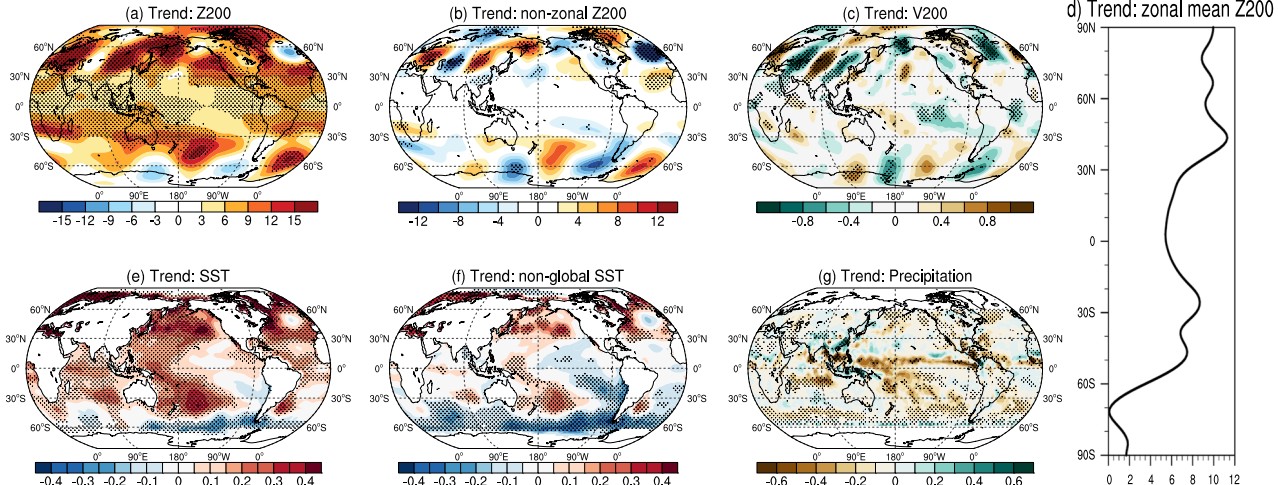

**Fig. 1 The recent increase in the wave amplitude of midlatitude circulation and concomitant global climate patterns.** Observed circulation and SST trends over the past 40 years: Linear trends of JJA **a** Z200 (unit: m/decade), **b** non-zonal Z200 (unit: m/decade), **c** V200 (unit: m/s/decade), **d** the zonal mean component of Z200 (unit: m/decade), **e** sea surface temperature (SST; unit: K/decade), **f** "non-global SST" (calculated as the SST pattern with the global mean SST removed in each grid; unit: K/decade), and **g** precipitation (unit: mm/day/decade) from 1979 to 2018 in observations. Z200 and V200 are the geopotential height and meridional wind at 200 hPa, respectively. The stippled areas indicate significant trends above the 95% confidence level.

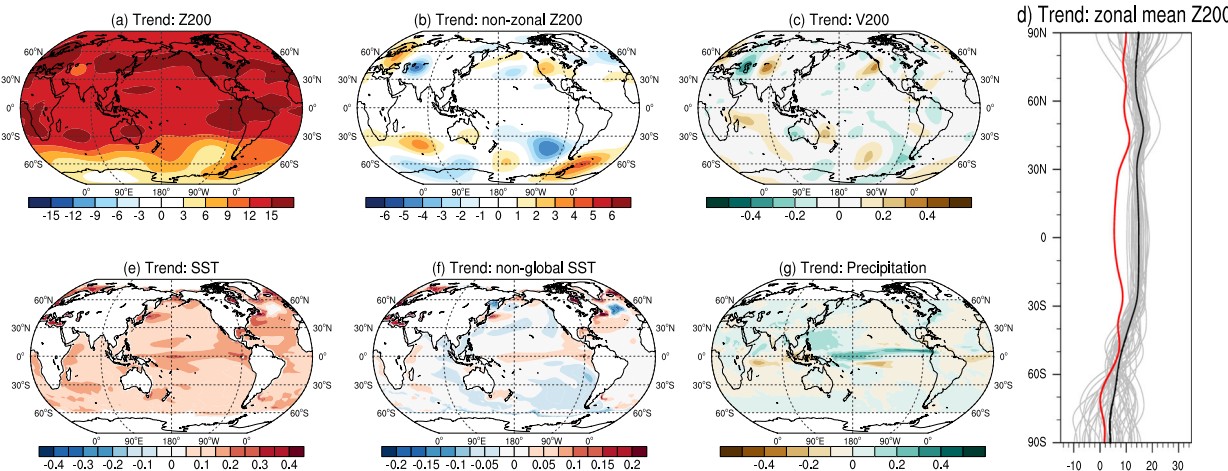

**Fig. 2 External forcing alone cannot explain the observed wave amplification.** Circulation and SST trends over the past 40 years from the ensemble mean of the CESM-LE historical run: linear trends of JJA **a** Z200 (unit: m/decade), **b** non-zonal Z200 (unit: m/decade), **c** V200 (unit: m/s/decade), **d** the zonal mean component of Z200 (unit: m/decade), **e** SST (unit: K/decade), **f** non-global SST with the global mean SST removed in each grid (unit: K/decade), and **g** precipitation (unit: mm/day/decade) from 1979 to 2018 in the ensemble mean of CESM-LE historical simulations. The gray lines in **d** represent each of the 40 members and the black line is the ensemble mean with the observed one (in red) derived from Fig. 1d.

Central Pacific (TWCP), along with a cooling SST trend in the TEP, may contribute to shaping the observed climate variabilities in the tropics and extratropics and the extent to which they are connected.

**The sensitivity of midlatitude circulation to tropical forcing.** To examine the role of suppressed convection in the TWCP in contributing to the observed circulation changes, we analyse the CESM1 pre-industrial (PI) simulation. One can better understand the behavior of Arctic and tropical climates in a modeling environment that is constructed to properly simulate the observed CGT-like circulation trend pattern in the extratropics. In doing so, we create a pseudo-ensemble with 1761 members by trimming the 1800-year-long CESM1 PI simulation into consecutive 40-year-long periods[45] (Methods section). In this way,

each pseudo-ensemble member corresponds to a 40-year-long period that is analogous to the past 40 years of observations in the absence of anthropogenic forcing, in which the long-term trend over the past 40 years, if present, is solely generated by internal variability of the CESM1. Then, we compute the spatial correlation between the observed (ERA5, 1979–2018) JJA non-zonal Z200 trend pattern (20°N–60°N, Fig. 1b) and the one in each member of the CESM1 pseudo-ensemble (Fig. 3). There is a prominent spread in the correlation coefficients with respect to each member (Fig. 3a, [−0.6, 0.7]) and the spatial correlations are characterized with recurrent positive and negative coefficients on low-frequency timescales. The members with significant positive ($r > 0.4$, 23 cases: positive group) or negative ($r < −0.4$, 16 cases: negative group) spatial correlation represent those periods exhibiting a similar extratropical short-wavy structure as observed,

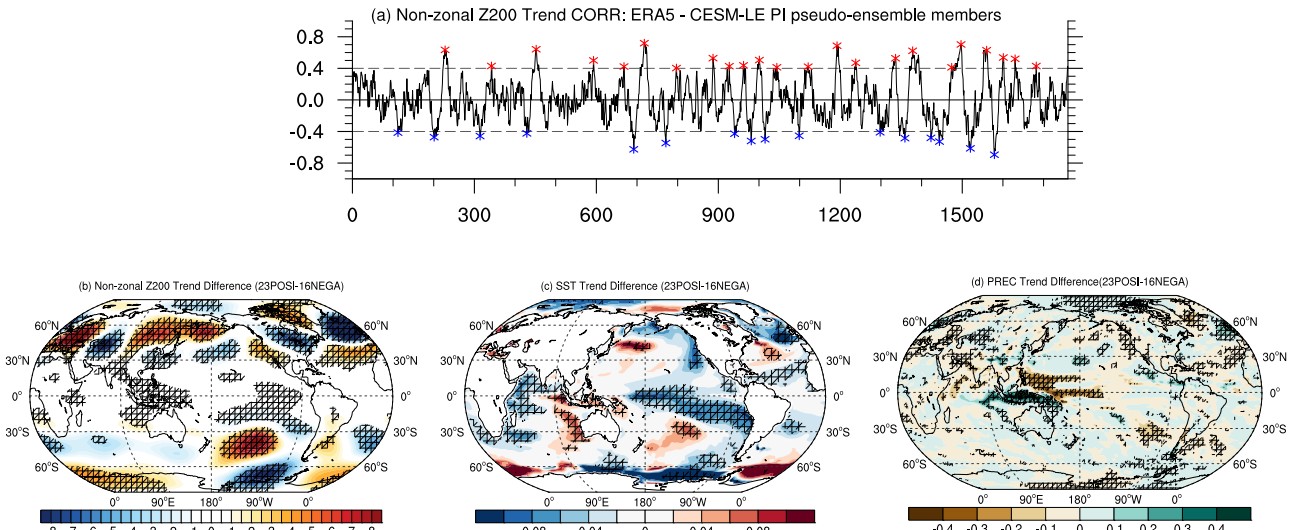

**Fig. 3 Simulated wavy pattern in the pseudo-ensemble of CESM-LE PI runs. a** spatial correlations (CORR) between non-zonal JJA Z200 trend derived from ERA5 over the past 40 years (1979-2018) and non-zonal Z200 trend over a 40-year period from each member of the pseudo-ensemble of CESM-LE PI simulation runs within the NH (20°N-60°N). The differences of **b** non-zonal Z200 (unit: m/decade), **c** SST (unit: K/decade), and **d** precipitation trends (PREC; unit: mm/day/decade) between the selected 23 positive (POSI) and 16 negative (NEGA) cases. The positive and negative cases (red and blue crosses in **a**) satisfy the following two conditions: (1) absolute values of the maximum and minimum spatial correlation coefficients are >0.4 and (2) the distances between any two adjacent members in each group (positive or negative) in the time axis are greater than 40 years. The cross-hatched areas in **b**–**d** denote the significant differences between 23 positive and 16 negative cases based on the two-sample t-test, p < 0.05.

either in-phase with or opposite to each other. Through making a difference of these two groups, we can examine the enhanced midlatitude waviness and its associated changes in the Arctic and tropics. The composite differences between the groups of positive and negative spatial correlations in Fig. 3b–d resemble many observed features in the midlatitude and tropics, including the wave train's position along the NH jet, the considerable SST cooling in the tropical Pacific, the anomalous convection in the TWCP with reduced rainfall along the ITCZ, and the enhanced rainfall over the Maritime Continent.

While these anomalous tropical SST and convection trends are similar to their observed counterparts over the past 40 years, the long-term climate anomalies in the Arctic show a weak cooling trend over the Russian Arctic as opposed to the observed warming in that region. Of these 23 cases, 9 cases (40%) show a slight warming trend in the Arctic and 17 cases (74%) exhibit a tropical SST cooling over the TEP (Supplementary Fig. 3). This suggests that the observed Arctic warming alongside the enhanced waviness in the midlatitudes is not a primary source for the generation of the CGT-like circulation trend in the simulations. Thus, the concurrence of the CGT-like circulation trend with the tropical SST and convection anomalies seen in both observations and the CESM1 PI simulation may represent a closely coupled mode on interdecadal time scales. Next, to understand the CGT-like circulation mode and its connections with Arctic variability, we construct an interannual Z200 Spatial Correlation Index (SCI) by calculating the spatial correlation between the JJA non-zonal Z200 trend in ERA5 (Fig. 1b) and simulated non-zonal Z200 in each summer of the CESM1 PI simulation within 20°N–60°N. The resultant SCI time series, characterized by pronounced low-frequency variability peaking around 30–50-year power spectrum bands, exhibits a close association with a cooling SST pattern over the tropical Eastern Pacific and suppressed rainfall over the TWCP but a weak connection with the Arctic climate anomalies (Supplementary Fig. 4).

We next examine the concurrence of the CGT and tropical convection anomalies in the CESM-LE historical simulation (Methods) to assess the sensitivity of the tropical forcing-CGT connection to anthropogenic warming. The spatial correlations between non-zonal Z200 trends over the NH midlatitudes (20-60°N) in each member of the CESM-LE (historical simulation) with their counterpart in ERA5 reveals that some members capture the observed changes better than others (Fig. 4a), even though the ensemble mean fails to replicate the observed wavy pattern (Fig. 2b). This infers that internal climate variability may play an important role in contributing to the spread of the simulated recent changes of the midlatitude circulations. To test whether those "good" members are able to capture the essential factors in the tropics with respect to the CGT-like circulation mode seen in the PI run, we compute the composite non-zonal Z200 and SST trend spatial patterns based on the difference between those 5 members showing the highest and lowest spatial correlations (20°N-60°N, 0°-360°) with observations, respectively (Fig. 4a). As shown in Fig. 4b, c, e, the composites closely resemble those seen in the observations and the PI simulation regarding the Z200, SST, and precipitation patterns. A TEP cooling with a weakened ITCZ precipitation and enhanced Maritime Continent precipitation paints a consistent picture with those associated with the CGT-relevant anomalies, in which Arctic temperatures are weakly yet predominantly cooler, not warmer (Fig. 4c).

This argument on the importance of suppressed tropical convection as a primary contributor to the observed midlatitude waviness over the past 40 years is reinforced by the scatter plot (Fig. 4d): members that have a better skill in capturing the CGT-like circulation are also better in replicating the TEP cooling and the suppressed TCWP precipitation (Fig. 4f), indicating that tropical forcing serves to generate the CGT-like wavy pattern more consistently than AA does. In contrast, a similar scatter plot with respect to the model's performance in replicating Arctic warming does not feature such a linear relationship. These results

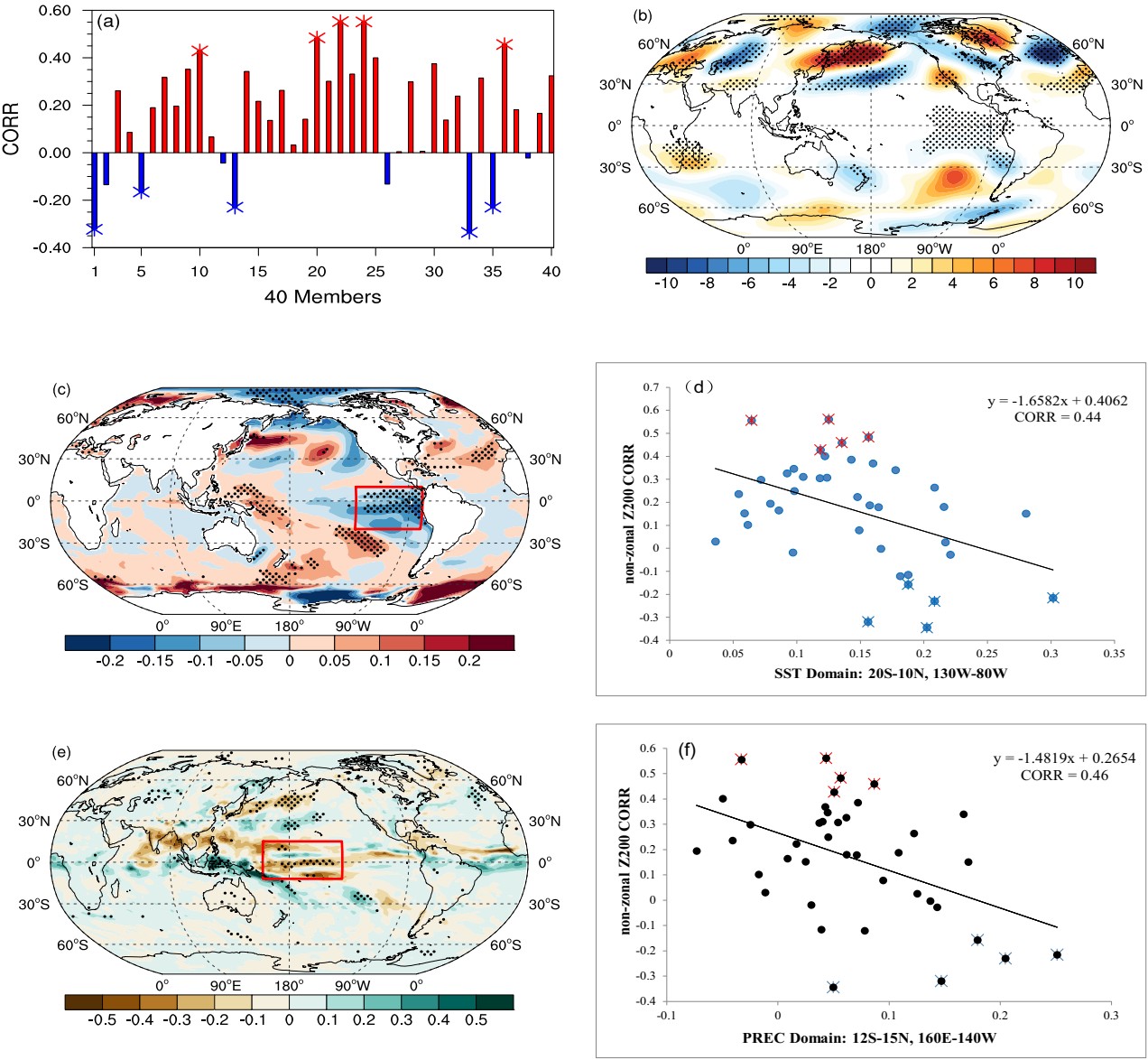

**Fig. 4 The circulation, SST, and precipitation trends over the past 40 years derived from CESM-LE 40-member historical simulation. a** spatial correlations between non-zonal JJA Z200 trend derived from ERA5 and CESM-LE 40-member historical simulation over the past 40 years (1979–2018) within the mid-high latitudes (20°N–60°N). The differences of **b** non-zonal Z200 (unit: m/decade), **c** SST (unit: K/decade), and **e** precipitation (unit: mm/day/decade) trends between the members with the five largest (red markers in **a**, **d**, and **f**) and five smallest (blue markers in **a**, **d**, and **f**) spatial correlations in **a**. **d** The scatter plot shows the relationship between domain-averaged SST in tropical eastern Pacific (20°S–10°N, 80°W–130°W) and spatial correlations in **a**. **f** The scatter plot shows the relationship between domain-averaged precipitation in the tropical West-Central Pacific (12°S–15°N, 160°E–140°W) and spatial correlations in **a**. The linear fitting lines and related correlation coefficients (CORR) are indicated in **d** and **f**, respectively. The stippled areas in **b**, **c**, and **e** denote the significant differences based on the two-sample t-test, $p < 0.05$.

reinforce our hypothesis that the observed CGT pattern in the Z200 trend, i.e., the pronounced short-wave structure along the westerly jet, is strongly linked to tropical forcing on both interannual and interdecadal timescales, regardless of the presence of anthropogenic forcing and AA.

To place the recent amplified wavy circulation pattern into the context of the long-term circulation trend in the observational data, we perform empirical orthogonal function (EOF) analysis (Supplementary Figs. 5 and 6) on the NH V200 (20°N–90°N) over the past 180 years using the NOAA 20th century reanalysis. The results suggest that the low-frequency variability of tropical SST and precipitation anomalies is closely associated with a CGT-like wave pattern over the longer period.

**The dynamical mechanisms contributing to midlatitude wave amplification.** In view of the tropical forcing as previously discussed, we propose two possible physical mechanisms to explain the enhanced midlatitude waviness. As discussed by Green's theory[46], the wave number of the stationary component of the zonal mean flow in the midlatitudes is controlled by the mean baroclinicity, which is sensitive to the equator-to-pole thermal gradient. Constrained by the thermal wind relationship, the waviness of quasi-stationary wave trains is enhanced when the meridional temperature gradient is reduced[4]. Previous studies mostly emphasized the role of anthropogenically excited AA in reducing the north-south temperature gradient over the past decades[47]. However, the tropical troposphere also experiences

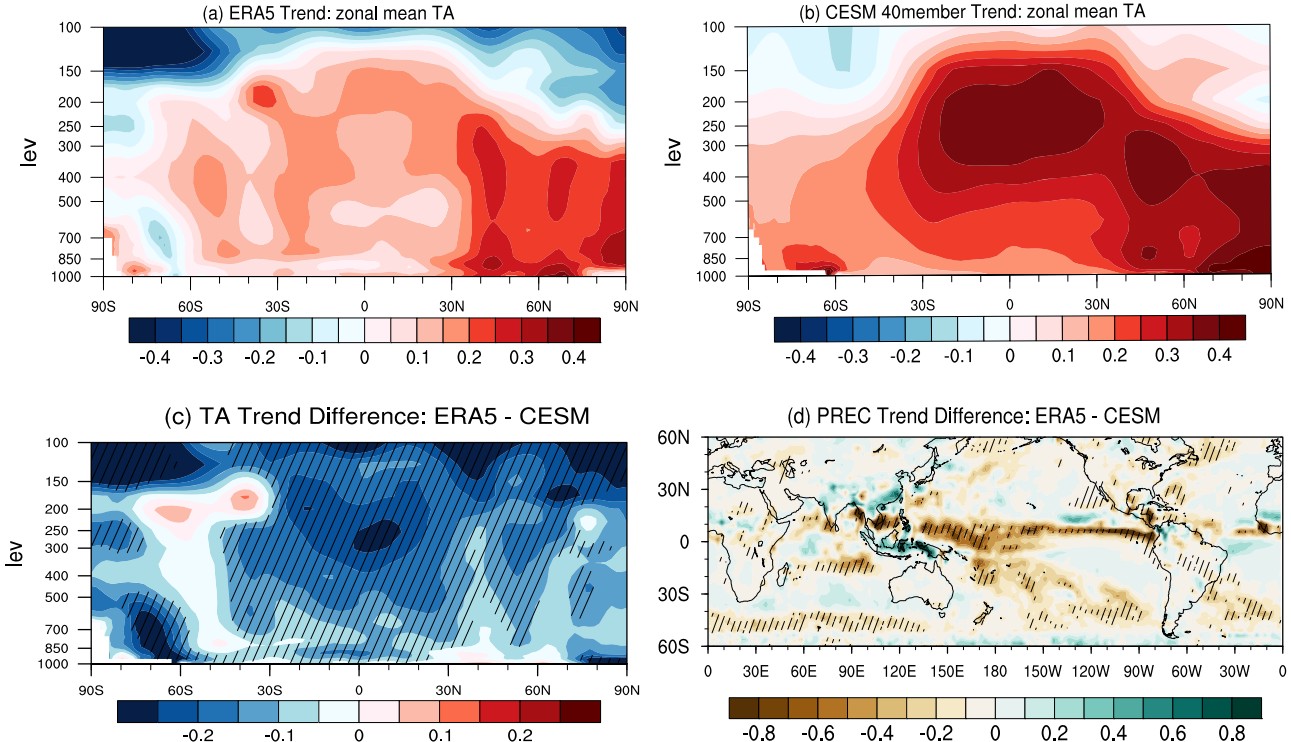

**Fig. 5 The stark contrast between ensemble mean simulated and observed vertical temperature and precipitation trends in the tropics.** JJA zonal mean component of air temperature (TA) trend profile from **a** ERA5 (1979-2018, unit: K/decade) and **b** the ensemble mean of CESM-LE 40-member runs (1979-2018, unit: K/decade), respectively. **c** Differences of zonal mean TA trends between ERA5 and the ensemble mean of CESM-LE 40-member historical runs. **d** Differences of JJA precipitation (PREC) trends (60°S-60°N; unit: mm/day/decade) between ERA5 and the ensemble mean of CESM-LE. The hatched areas in **c** and **d** indicate the ERA5 trends lie outside two standard deviations away from the mean of the CESM-LE 40-member simulations.

significant warming under the constraint of the moist adiabatic lapse rate. Thus, anthropogenic forcing and tropical climate variability induce a tug-of-war scenario between the enhanced warmings in the tropics and the Arctic, affecting the stability of the north-south temperature gradient[25].

Through the diagnostics in this study, we find that all models overestimate the tropospheric warming trend over the past four decades at almost all latitudes, particularly in the tropics (Fig. 5a±c and Supplementary Fig. 7a–d). In response to increasing anthropogenic forcing over the past four decades, the ensemble means of CESM-LE, CMIP5, and CMIP6 produce stronger tropospheric warming trends and enhanced precipitation over the TWCP than in observations (Fig. 5c, d and Supplementary Fig. 7c–f), which makes the long-term change of the tropospheric tropical-Arctic temperature gradient in models behave differently from that in reality. This difference may partially result from the prevalence of SST cooling and suppressed convection along the ITCZ over the tropical Pacific in the observations, which can additionally weaken the temperature gradient between the tropics and the Arctic further than if solely forced by $CO_2$. Consequently, models forced by greenhouse gases alone can only capture parts of the observed waviness along the midlatitudes (Fig. 2). We thus suggest that the observed tropical SST cooling and associated suppression of tropical convection during the past decades may act as an additional mechanism, if not primary, along with the $CO_2$-induced AA, to contribute to the recent enhancement in the waviness of midlatitude jet stream through the reduction of the equator-to-pole temperature gradient.

It is known that tropical convection anomalies alone can enhance the waviness in the extratropical circulations[48] in a manner that resembles the CGT. In summer, the CGT represents the most unstable mode of the extratropical mean flow owing to

its sensitivity to the barotropic instability of the basic flow, which is most active over the jet exit region over the Northeast Atlantic[49]. To illustrate this process, we calculate the standard deviation of non-zonal Z200 in the three reanalyses and CESM-LE control simulation (Supplementary Figs. 8a–b and 9a–b). The most prominent center of variability is located over the northeast Atlantic jet exit region (45°N-55°N; 0°-20°W). Next, we calculate the correlation maps of the non-zonal Z200 over the jet exit region with the global SST (Supplementary Figs. 8c–d and 9c–d) and precipitation fields in three reanalyses and CESM-LE (Supplementary Figs. 8e–f and 9e–f). The correlation maps show that negative SST anomalies in the TEP and positive precipitation in the Maritime Continent are associated with a low-pressure anomaly in the Northeast Atlantic. Similar correlation patterns are found in ERA-20C, NOAA-20CR, and CESM-LE simulation. These robust linkages manifest as the effect of tropical forcing in triggering strong barotropic instability over the jet exit in the Northeast Atlantic[49].

To explore the mechanisms of this linkage, we perform two sets of experiments that include 10 members of control and sensitivity experiments each (Methods section). The control run is a typical PI simulation with anthropogenic forcing set at constant values. In the sensitivity simulations, a pair of diabatic heating anomalies is imposed in the tropics to imitate tropical forcing based on the long-term precipitation changes over the past 40 years (Fig. 1g). The JJA Z200 differences between the ensemble average of the sensitivity and control runs (Fig. 6a) illustrate a clear wave train structure bearing resemblance to the observed CGT, and the spatial correlation of this response with the observed Z200 trend pattern is 0.68 in the extratropics (20°N-60°N, 0°-360°). The strong low-pressure center generated over the Northeast Atlantic suggests that the added tropical forcing enhances barotropic instability at the jet exit region[50].

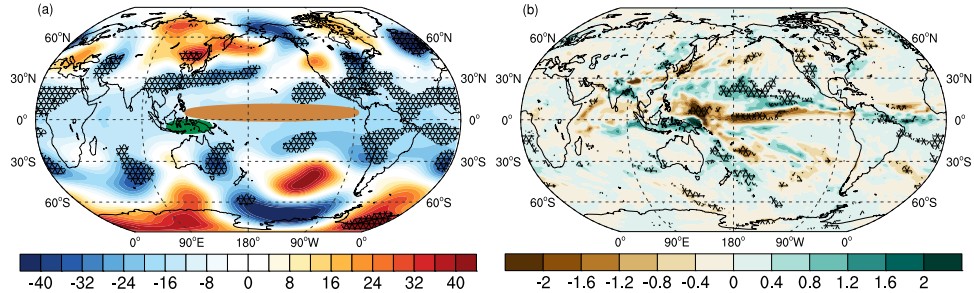

**Fig. 6 Simulated global JJA Z200 and precipitation response to imposed diabatic heating anomalies in the tropics. a** The Z200 (unit: m) and **b** precipitation responses (mm/day) in the CESM1 to two additional heating sources added in tropical Western and Central Pacific. The green (10°S-0°, 110°E-150°E) and brown (0°-10°N, 120°E-80°W) color-filled ovals imposed in **a** denote the location where a pair of positive and negative heating sources are added in the CESM1. The cross-hatched areas in **a** and **b** denote the significant differences between the CTL and SEN experiments (Methods section) based on the two-sample *t*-test, $p < 0.05$.

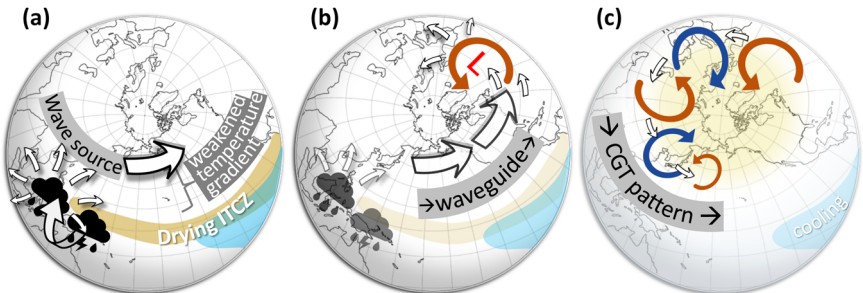

**Fig. 7 A schematic diagram illustrating the mechanism.** The mechanism is comprised of three steps: Suppressed ITCZ convection and enhanced convection over the Maritime Continent trigger the formation of the CGT-like short-waves pattern along the jet stream (and other higher-latitude wave trains) by exciting strong local RWS over the Western North Pacific and weakening the pole-to-equator temperature gradient (step **a**); These changes will further trigger barotropic instability around the jet exit (step **b**) over the Northeast Atlantic and the establishment of the CGT over Eurasia (step **c**). See the main text for a more detailed description of the mechanism.

The tropical rainfall response (Fig. 6b) to the specified diabatic heating looks similar to the observed precipitation trend over the past 40 years. Since these imposed heating will trigger complex responses of processes associated with cloud physics, vertical diffusion, and radiation in the tropics, the overall diabatic heating to determine the extratropical response is a combination of imposed and induced diabatic heating anomalies in the model. A resemblance between the simulated rainfall response and observed counterpart indicates that our imposed heating can trigger reasonable tropical heat source as what has happened in the real world. Furthermore, this imposed tropical forcing induces a tropical-wide cooling (Fig. 6a) that is conducive to the reduction of the equator-to-pole thermal gradient. For diagnostic purposes, we perform two additional sensitivity experiments to examine the importance of individual heating source in driving the extratropical circulation and find that the ITCZ cooling is more critical than heating over the Maritime Continent in creating the midlatitude wave-train structure. Taken together, these experiments lend further support that tropical convection anomalies, which feature a suppressed convection along the ITCZ and an enhanced convection over the Maritime Continent, play a key role in increasing the waviness along the jet through the formation of the CGT and the weakening of meridional temperature gradient.

## Discussion

In all, the tropical processes leading to the waviness in the Z200 trends (Fig. 1) are manifold and can be summarized in a schematic diagram (Fig. 7a): first, the recent SST and precipitation trends in the tropical Pacific and its associated upper-level convergence/divergence anomalies (Supplementary Fig. 10) generate

strong Rossby wave source (RWS, Methods) over the Western North Pacific, which in turn creates propagating Rossby waves at upper levels and subsequently disturbs the mean flow (Fig. 7a). This mechanism is supported by the observed and simulated RWS in Supplementary Fig. 10. Then, the westerly jet acts as a waveguide for the perturbation to propagate downstream from the Pacific to the North Atlantic, while the excited disturbance sends off wave energy that is accumulated at the jet exit region over the Northeast Atlantic, which is manifested as a low-pressure center (Fig. 7b). Once low pressure is established there, wave energy tends to propagate southward following an arch-shape pathway to reach central Asia and then continues to propagate along the subtropical jet (Fig. 7c) with a particular wavelength (~70 degrees) that is determined by the mean flow[51]. These conditions favor the formation of the negative-phase CGT. Although the barotropic instability of the mean flow (i.e. horizontal wind shear) can amplify the CGT, the presence of low-frequency diabatic forcing in the tropical Pacific appears to trigger and maintain its establishment, hence imprinting the CGT's distinct spatial feature onto the extratropical circulation trend over the past 40 years. Meanwhile, the overall suppressed convection along the ITCZ favors a weakening of the equator-to-pole temperature gradient that can further enhance the waviness of the jet through increasing the zonal wavenumber of trapped stationary Rossby waves (Fig. 7a)[24]. We should note that other types of higher-latitude wave trains could also interact with the CGT[52] and contribute to the establishment of the circulation trends. Future analysis to better understand the possible interactions of these cross-latitude wave trains is needed. The next important question concerns the origin of the cooling TEP, yet current climate models lack the critical performance to establish a

reliable attribution analysis[41,53]. Considering that the enhanced waviness occurring in tandem with a tropical SST cooling can be detected in the PI simulation, we believe that the enhanced midlatitude waviness may be of internal origin, though we cannot rule out the possibility that anthropogenic forcing may also force the enhanced waviness through generating a cooling trend of SST in the TEP.

Considering that the enhanced waviness may modulate the occurrence of extreme weather events in the midlatitudes, it is arguable that the widespread increases in extreme weather in the extratropics is partially attributable to the heating change in the tropical Pacific. Thus, projections of extreme events along the westerly jet may not only depend on the climate response to anthropogenic forcing in the Arctic but also are subject to tropical variability that are sensitive to both internal and anthropogenic forcing. Quantification of the relative roles of this tropical variability and the AA effect in shaping the recent circulation changes requires further attention[54–57]. Likewise, the prediction of low-frequency SST variability in the tropics is a crucial future work in order to reduce the uncertainty associated with projecting midlatitude extreme weather.

## Methods

**Reanalysis and SST data**. We use monthly mean atmospheric variables from the European Center for Medium Range Weather Forecasting (ECMWF) reanalysis ERA5 for 1979–2018[58] and longer monthly data from ECMWF's ERA-20th Century (ERA-20C) reanalysis[59] for 1900–2010 and the NOAA 20th Century Reanalysis[60] (NOAA-20CR) for 1836–2015. The raw datasets are regridded onto the 1.5° × 1.5° regular grid to facilitate a comparison among different data sources. Monthly sea surface temperature (SST) data is obtained from NOAA Extended Reconstructed SST (ERSST) version 5[61], with a resolution of 2° × 2° for 1854–2018. We also use ECMWF's ERA5.1, a re-run of ERA5 for 2000-2006, to re-evaluate the long-term trend of zonal mean air temperature in JJA (Supplementary Fig. 11). ERA5.1 combined with ERA5 shows a slightly enhanced warming trend from the upper troposphere to lower stratosphere (above 300 hPa) and a weaker warming trend in the lower troposphere (below 800 hPa). Overall, the new temperature trend reflected by a combination of ERA5.1 and ERA5 is similar to that using ERA5-only in the troposphere, and we thus only use ERA5 in our calculations.

In this study, non-zonal Z200 denotes the Z200 for each grid minus the zonal mean component of Z200, and non-global SST denotes the SST at each grid minus the mean SST of 60°S-60°N. The SST data used from ERSSTv5 in Fig. 1f is JJA non-global SST. Figure 3c uses the raw JJA SST data derived from CESM1 1800-year PI control run. The raw JJA SST data derived from CESM1 40-member large ensemble forced run are utilized in Fig. 4c.

**Model simulations**. In order to explore the drivers of observed extratropical circulation variability in the most recent 40 years, we utilize the Community Earth System Model1- Large Ensemble (CESM-LE)[62] project that is designed to distinguish the influence of internal climate variability and external forcing on the climate system. Furthermore, the 1800-year-long CESM1 PI control simulation enables us to assess the contribution of internal climate variability to recent circulation variability in the absence of anthropogenic forcing. We also utilize the CESM1 40-member large ensemble forced by historical and RCP8.5 external forcing scenario, which allow us to evaluate the models' responses under global warming conditions from 1979 to 2018. In addition, historical(1979-2005) +RCP8.5 simulations (2006–2018)[63] of 30 climate models from CMIP5 and 35 models with historical forcing (1979–2014) from CMIP6[64] are used to examine whether features disclosed in observations can be found in other models in the community archived in CMIP5 and CMIP6 (Supplementary Table 1). We mainly focus on the ensemble mean of CMIP5 and CMIP6 models to reduce uncertainty of the differing model physics. All model outputs are regridded onto the 1.5° regular grid before analyzing the model results.

To make good use of other available CESM1 experiments conducted by NCAR Climate Variability & Change Working Group, we also examine the ensemble means of an Atmospheric Model Intercomparison Project[65] (AMIP) type Tropical Ocean Global Atmosphere[66] (TOGA) experiment and a Pacemaker run[62], in which observed SST anomalies are nudged within the tropical Pacific (10°S–10°N, 160°W–90°W). Historical anthropogenic forcing is imposed in both these two experiments. The rainfall trends over the Western North Pacific in the TOGA run over the period (1979-2005 of historical experiments and 2006-2017 of RCP8.5 simulations) are nearly opposite to the observed pattern since the observed rainfall change results from active atmosphere-ocean coupling in the tropics rather than a sole response of the atmosphere to SST forcing as designed in AMIP runs[67]. Without a correct simulation of the tropical rainfall trend pattern, it is not very informative to further examine the response in the extratropics in the TOGA run.

As we expect, the Pacemaker run captures a better rainfall trend as observed over the tropical Pacific than the TOGA run and CESM-LE. The extratropical responses are also moderately improved compared with that in CESM-LE (Supplementary Fig. 12), indicating the importance of tropical convection variability in contributing to a better simulation of midlatitude circulation trends over the past 40 years.

**Pseudo-ensemble sampling**. The main goal of this approach is to search for 40-year-long periods within simulations that exhibits strong similarity to observed circulation trends. To do so, Z200 and SST during consecutive 40-year periods are used to create a pseudo-ensemble[45,68] out of long PI integrations. Therefore, we create a pseudo-ensemble with 1761 members and each member corresponds to a 40-year-long time-series. This pseudo-ensemble is used as an analog of the given model's internal climate physics in the absence of anthropogenic forcing. We then search for the 40-year-long periods showing the most similar Z200 pattern based on linear trends of JJA non-zonal Z200 in ERA5.

**A sensitivity experiment examining the extratropical response to imposed tropical diabatic heating anomalies**. To achieve this goal, we use the CESM1[69] to perform two sets of simulations including control (CTL) and sensitivity (SEN) experiments. CTL contains 10 members of 1 year simulations from January 1 to December 31 initialized from different initial states on January 1 with the same and fixed anthropogenic forcing. SEN has the same configuration (e.g. initial states, ensemble members, and anthropogenic forcing) as CTL except that in the tropics a pair of diabatic heating anomalies are imposed throughout the 12-month integration period. Imposed heating includes a cooling patch placed along the ITCZ (0°-10°N, 120°E-80°W) and a positive one centered over the Maritime Continent (10°S-0°, 110°E-150°E). This pattern imitates the key feature of the long-term trend in observed rainfall in the deep tropics over the past 40 years (Fig. 1g), which is believed to be critical to enhanced waviness in the extratropics.

To determine how much diabatic heating rate should be added in the model, we calculate the rate based on a relationship between condensation and related latent heat release. Namely, a 1-mm/day rainfall anomaly is equivalent to a uniform 0.25 K/day heating rate throughout the air column from the surface (1000 hPa) to the top of atmosphere (the air mass in the whole air column is 1000 hPa). However, in reality, the diabatic heating is mostly released in the middle of the troposphere and this heat is primarily used to warm the air column below the tropopause (200 hPa; the air mass to be heated is 800 hPa). In addition, in the horizontal dimension, the heating is maximized at the center of a patch and small around the margin. By considering all these factors, 1 mm/day rainfall anomaly corresponds to about a maximum of 1.2 K/day heating rate in our idealized heating structure. Since the observed rainfall trend in the tropics has maximum values ~0.4–0.5/mm/day/decade along the ITCZ and Maritime continent, based on the simple scaling relationship, we estimate that the maximum heating rate at 500 hPa should be around 0.5 K/day. So, in the SEN experiment, the heating rates for both the positive and negative heating forcings have the maximum values at 500 hPa and are linearly deceased to zero at 200 hPa and the surface in the vertical direction. In the horizontal plane the heating rates have the maximum values at the center of each patch (0.5 °C/day and −0.5 °C/day) and are linearly decreased to zero at the edges. The difference of 10-member mean of SEN and CTL in JJA are used to estimate how tropical forcing drives the midlatitude wave amplification and the significances of the difference are assessed by the two-sample T test.

**Rossby wave source RWS**. The RWS is calculated to understand how tropical forcing contribute to the formation of the CGT. According the derivation of Sardeshmukh and Hoskins[48] (1988) on the basis of the nonlinear vorticity equation, the idealized Rossby wave source (RWS) at a certain level can be calculated as:

$$RWS = -\nabla \bullet (V_\chi \zeta) \tag{1}$$

where $\zeta$ is absolute vorticity calculated as the sum of relative vorticity and planetary vorticity. $V_\chi$ is divergent component of the total wind V.

**Statistical tools and significance**. The main statistical tools utilized in this study include empirical orthogonal function (EOF) analysis, Fourier power spectral analysis, and composite analysis. To identify whether the CGT-like pattern represents a leading circulation mode of mid-latitude circulation over a long period, we use EOF analysis to extract the orthogonal leading modes of V200 in the NH (Supplementary Fig. 5a–c). Fourier power spectral analysis is further used to reveal the periodicity of the principal component of this mode (Supplementary Fig. 6b). The EOF2 of meridional winds at 200hPa in 3 reanalysis datasets can consistently capture a CGT-like pattern (Supplementary Fig. 5d–f) and exhibits its strong interdecadal variations (~30–50-year). The method examining the significance of correlation coefficients in this study is to modify N with an "effective sample size" N* by calculating the autocorrelation of two variables[70]:

$$N^* = N \frac{1 - r_1 r_2}{1 + r_1 r_2} \tag{2}$$

where $r_1$ and $r_2$ are lag-one autocorrelation coefficients of each variable and $N$ is the original sample size. A 95% confidence level is used to determine the significance of the correlation coefficients.

## Data availability

All reanalysis data used in this study were obtained from publicly available sources: ERA5 reanalysis data can be obtained from the ECMWF public datasets web interface (http://apps.ecmwf.int/datasets/). The ECMWF ERA5.1 reanalysis product is available at https://confluence.ecmwf.int/pages/viewpage.action?pageId=181130838. Simulated global circulation, temperature and sea ice under anthropogenic forcing were obtained from the CMIP5/6 and CESM-LE archives accessed through the Earth System Grid Federation data portal (http://esgf.llnl.gov). The data generated in this study has been deposited in the OSF and can be accessed via https://osf.io/xq9ap/. In addition, the data generated for this paper and the CESM1 heating-imposed experiment raw output is available from the corresponding author upon request.

## Code availability

Previous and current CESM1 versions are freely available at www.cesm.ucar.edu:/models/cesm2/. The data in this study are analysed with MATLAB. Contact Q.D. for specific code requests.

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

## Acknowledgements
This study was jointly supported by Qingquan Li's grants (the Strategic Priority Research Program of the Chinese Academy of Sciences (Grant No. XDA20100304), the Second Tibetan Plateau Scientific Expedition and Research Program of China (Grant No. 2019QZKK0208), the National Natural science Foundation of China (Grant No.41790471), Modelling, Analysis, Predictions and Projections (NA19OAR4310281) and Climate Variability & Predictability (NA18OAR4310424) programs as part of NOAA's Climate Program Office, and NSF's Polar Programs (OPP-1744598). S.Y.S.W. was supported by the U.S. Department of Energy under Award Number DE-SC0016605 and by the National Science Foundation P2C2 Program under Award Number 1903721.
D.T. was supported by the Doctoral Student Scholarship Program of the Co-operative Doctoral Program of the Ministry of Innovation and Technology financed from the National Research, Development and Innovation fund. H.T. was supported by the Office of Science, U.S. Department of Energy Biological and Environmental Research as part of the Regional and Global Model Analysis program area. The Pacific Northwest National Laboratory (PNNL) is operated for DOE by Battelle Memorial Institute under contract DE-AC05-76RLO1803. We acknowledge the CESM Large Ensemble Community Project led by Climate Variability & Change Working Group, and supercomputing resources provided by NSF/CISL/Yellowstone (https://doi.org/10.5065/D6RX99HX) and CESM Polar Climate Working Group.

## Author contributions
Q.D. and Q.L. led this work with contributions from all authors. Q.D. conducted the experiments. X.S. made the calculations and created the figures. Q.D, S.Y.S.W., and X.S. led analyses, interpreting results, and writing the paper. All (X.S., Q.D., S.Y.S.W., D.T., Q.L., C.C., H.T., R.L., and Y.D.) authors discussed the results and contributed to writing the paper.

## Competing interests
The authors declare no competing interests.
