## [Peer Review File · Nature Communications]

Enhanced jet stream waviness induced by suppressed tropical Pacific convection during boreal summerREVIEWER COMMENTS

Reviewer #1 (Remarks to the Author):

The paper explores possible reasons of wavier jet in the Northern Hemisphere, that has been observed in the recent decades. Model simulations show that under modified tropical conditions in the Pacific + Maritime continent, wavier jet is found in a few simulations. The authors conclude that this forcing may be responsible for the NH wavier jet, rather than the Arctic Amplification, that had been suggested as the key reason by some previous studies.

I like the clarity of the manuscript in setting-up the research question. I have no doubts that it falls within the scope of Nature Communications. However, there are a few major comments that need to be addressed before the manuscript can be considered for publication:

1. In the introduction, the authors state that there are two 'schools of thinking' on whether the wavier jet is caused by the Arctic Amplification (AA) or the other way round, i.e., wavier jet causes a warmer Arctic. While there are papers that, indeed, make a statement that the warmer Arctic is responsible for the wavier jet, the other statement remains a speculation, not a finding (Blackport and Screen (2020) explicitly say that they 'speculate'). While I can see how wavier jet can induce short-term temperature anomalies in the polar regions, I don't think anyone has been able to show that this is the mechanism of the recent trend in the Arctic.

Furthermore, this manuscript shows that simulations with waiver jet have a cooler Arctic, not warmer. Hence, I believe that it is plausible to link a wavier jet to a tropical forcing, but connection to the AA is not supported by findings of this paper.

2. The mechanism through which suppressed ITCZ convection and enhanced convection over the Maritime Continent trigger the formation of the CGT-like short-waves pattern along the jet stream suggested in the paper should be better supported by the results.

Firstly, no figure in the main manuscript shows any wave characteristics. Fig. 10 in the supplemental information shows the wave activity flux along with divergent wind component and velocity potential. This mechanism explaining the importance of the tropical forcing is a key finding and, hence, I believe that Sup. fig. 10 should be moved into the paper.

But more importantly, I do not see how Sup. fig. 10 supports the mechanism offered in Fig. 6: there is no wave activity flux coming out of the Pacific or Maritime Continent (marked as 'wave source' region in Fig.6), neither in the observational nor in modelled data. I would like to see a plot that actually supports the proposed mechanism.

3. Lastly, a more detailed description of methods is needed (in lines 379-386). It is not clear to me how the correlation between fields was evaluated. What happened after the EOF analysis? What was the area that was compared? Did you estimate correlation between the PC2 (as PC2 is shown in Supplements, but it was not mentioned in the main manuscript) of the observed and modelled V200, their low-frequency components or something else? What are the two variables in eq. at line 383?

Other comments:

l.98: This drying trend at 5N has been linked to a northward displacement of the ITCZ due to stronger warming of the NH, see, e.g.:

Schneider, T., Bischoff, T. & Haug, G. Migrations and dynamics of the intertropical convergence zone. *Nature* 513, 45–53 (2014). <https://doi.org/10.1038/nature13636>

High-latitude eruptions impacts on ENSO and AMOC

Francesco S. R. Pausatà, Leon Chafik, Rodrigo Caballero, David S. Battisti

Proceedings of the National Academy of Sciences Nov 2015, 112 (45) 13784-13788; DOI:

10.1073/pnas.1509153112

I.186: I don't understand this statement. Recent Arctic SIC decline would be not captured if SIC variability below 30 years is filtered out.

I. 193: why Sup. Fig. 4 is not in the manuscript? This section refers to it a lot (at least five times), it would be easier to read if the figure was added to the paper.

I.198: Is it global/hemispheric trend in Z200 /SST or in selected regions?

I.204: See my major Comment 1. Were you able to find any simulations where wavier jet was associated with a cooler tropical Eastern Pacific and a warmer Arctic at the same time?

I.299: can this wavelength be calculated?

Minor:

I.89: almost everywhere in the NH, but not the SH, where the trend pattern is more complex.

I.149: the long term trend over the past 40 years, if present, is ...

I.168: Why 'subtle'? perhaps, replace with 'some'

I.242: awkward wording

I. 271, 301, ... : fix 'diabetic' heating (I saw it 4-5 times throughout the paper)

Reviewer #2 (Remarks to the Author):

Enhanced jet stream waviness induced by suppressed tropical Pacific convection during boreal summer

This paper examines the potential role of suppressed convection in the western tropical Pacific in driving trends in the extratropical circulation over recent decades. Previous studies have argued for the potential role of Arctic amplification in these "waviness" trends and this study puts forth an alternative hypothesis and argues that internally generated SST trends may have been responsible through their influence on tropical convection, wave generation and downstream teleconnections through the circumglobal waveguide. The arguments have been backed up by analysis of observed trends, internal variability in model simulations and in idealized experiments in which heating perturbations are imposed in the tropics within a GCM. Overall, I think this study is interesting and relevant and should be worthy of publication. However, I do have a number of comments that I recommend be considered before publication. While I don't see any of these as being particularly major, I think I have a sufficient number of them that, taken together, they amount to major revisions.

General comments:

(1) Tightening up the arguments for whether or not the SST trend is internally generated: Currently, I think the authors may have been too quick to come to the conclusion that the observed SST trend is internal. This is stated throughout the text, and I don't think it has been adequately acknowledged that there is actually considerable debate over whether the cooling trend in the tropical Pacific may actually be anthropogenically forced and the models are doing something wrong (Seager et al 2019, Nature Climate Change, doi:10.1038/s41558-019-0505-x). I think that, given this debate, it's currently not ok to presume that the SST trends are internally generated. I think a valuable addition, would be to show whether the same magnitude of tropical

Pacific SST cooling trends can be found in any members of the piControl pseudo ensemble and the historical ensemble. I recommend that the authors show that the real world tropical Pacific SST trend is not outside of the distribution from the model ensembles, before making the argument that it is internal variability. But even then, I think it should still be acknowledged that there is debate over whether the SST trends are forced or internal.

(2) Being careful about statements regarding whether models are wrong or not and whether they can capture the observations. In the paragraph at l101 the observations are compared with the ensemble mean of the models and I think it could be better articulated what we get from this. I find the text to be a bit muddled as to whether it's saying the models are wrong or whether it's saying that the forced response does not resemble the real world trends. I think that latter is the conclusion that should be drawn but the use of statements like "CMIP6 exhibits some improvement" (l110) confuses matters because it makes it sound like we should be expecting the models to look like the real world. Then in the subsequent paragraph it refers to "These biases lead to the incorrect simulation...". This is very confusing. What biases? And if the claim that is made in the abstract is that the tropical Pacific SST trends are internal variability then we should not expect to see it in the CMIP ensemble mean which averages out the internal variability and shows the model simulation of the forced trend. I think this whole discussion needs to be improved. If you want to make arguments that what has been seen in the real world is consistent with the models and internal variability, then it needs to be shown that the ensemble spread encompasses the real world. Likewise, if you want to argue that the models are biased, it needs to be shown that the real world lies outside of the ensemble spread. I realize this has been done for the Z200 trends, but it hasn't been for everything else.

(3) I'd recommend some improved discussion of the relative magnitudes of aspects discussed in the figures. Wildly different color bars are used for the obs and the models but from eyeballing it, it does look like the magnitude of e.g., the SST anomalies relative to the Z200 anomalies is quite comparable. I recommend you do what you can to try to make the color bars similar to one another, but I understand that may not be possible, so I'd recommend at least adding some discussion about the relative magnitudes and whether they match up or not.

(4) An obvious analysis to perform that would back up the arguments made in the paper is to consider AMIP-type simulations or pacemaker simulations and I'm not sure why this hasn't been done. A suite of relevant experiments accompany the CESM1 large ensemble http://www.cesm.ucar.edu/working_groups/CVC/simulations/cam5-prescribed_sst.html. They may stop short of the exact years that are used in this analysis, but there are equivalent simulations available with CESM2 as well, that extend to present http://www.cesm.ucar.edu/working_groups/CVC/simulations/cam6-prescribed_sst.html. If the authors arguments are correct and the tropical Pacific trends are a primary influence on the circulation trends, then shouldn't we expect to see these trends in prescribed SST simulations? A comparison between the coupled runs and either GOGA or TOGA simulations seems like an obvious comparison to make and I'm wondering why the authors haven't done this.

(5) All the figures were kind of fuzzy in the pdf version I had and I think it's strange in Figs 1 and 2 to have panel (c) to the right of (d). Suggest reordering the panel labelling.

Comments by line number:

l33: At the outset here in the abstract it may not be clear to readers what "tropical internal cooling" is. I realize you may be up against a word limit but I'd recommend "for the role of internally generated tropical cooling"

l41: It's not clear what "jet persistency" means or how it really relates to the findings. Perhaps "jet waviness" would be more appropriate?

l43: I think references are needed for these studies that have "expressed concern about the increasing meandering". Perhaps it's some of the references mentioned in the following sentence, but I think it would be appropriate to cite the relevant ones here.

I47: "enhanced amplitudes" of what? stationary waves?

I59: "AA amplifies midlatitude stationary waves via the reduction of the equator-to-pole thermal gradient". I'm not sure that this is anchored in theory discussed in reference 19. I haven't read through the whole of that paper before making this comment, but I did check the "Tropospheric dynamics" section and I think it's arguing the opposite. It's arguing that an increased equator-to-pole temperature gradient will increase the baroclinicity and, therefore, the growth of atmospheric eddies. What the authors are referring to may be addressed in another section of ref 19 that I didn't read carefully, but I suggest they double check. In any case, I think it should also be acknowledged that the theory behind the enhanced waviness in association with arctic amplification is not at all clear cut, as discussed by Woollings and Hoskins (2015) doi:10.1007/s40641-015-0020-8.

I94: suggest "widespread cooling"  "widespread relative cooling" or "widespread cooling relative to the global average", since a cooling is not really apparent in Fig 1e.

I168: "subtle importance" is quite ambiguous. It sounds like you're arguing it is important but it's subtle. But what I think you're trying to say that it's not really important or maybe something like it "plays a smaller role compared to the relative cooling of the tropical Pacific"

I172: This section gets a bit frustrating with the continued reference to the supplementary material. If I understand correctly, the difference between Fig 3 and supplementary Fig 3 is that Fig 3 is for 40y trends and Fig S3 is for interannual variability. It may be less frustrating for readers if Supplementary Fig 3 is not given so much of a write up and is instead mentioned in the paragraph above with a simple statement that similar patterns are found when considering interannual variability and with some mention of the power spectrum. The same kind of goes for the paragraph at I188. It's described as a whole new analysis but we're only pointed toward supplementary figures, which gets frustrating, but really this is just confirmation that the same thing is seen in the historical simulations as in the piControl.

I217: It's not clear what "enhanced atmospheric variability" means here. I think instead of "enhanced atmospheric variability" you perhaps mean "atmospheric circulation trends"?

I280: I think it's an overstatement to say that the tropical forcing is "essential" to generating this wave train structure. Given the arguments related to barotropic instability of the jet exit region in the Atlantic, it seems likely that there are many different perturbations that could end up giving this structure if it is the way in which the mean flow is unstable e.g., Simmons et al (1983), JAS, 40, 6, 1363--1392). Suggest rewording.

I327: I'm not sure that the reference to Fig 6c is correct here since that is the schematic.

I330: Fig 4c doesn't show SSTs, it shows atmospheric temperature.

I358: Recommend being more specific about whether this is CESM1 or CESM2 and adding a reference.

I381: I appreciate the consideration of the temporal degrees of freedom, but I wonder if also the spatial degrees of freedom should be accounted for by considering the False Discovery Rate (Wilks (2014) "The stippling shows statistically significant grid points", BAMS)

Figure 4 c and d. I think it would be useful here to have stippling or shading to indicate where the reanalysis trends lie outside of the ensemble distribution of trends.

Figure 4: For the discussion surrounding this figure, I think there needs to be some acknowledgement that the overall diabatic heating perturbation that the model is seeing is the combination of that due to the precipitation anomalies plus the heating that is being imposed. We don't have a good sense of what the magnitude of that imposed heating perturbation is. I recommend showing or discussing that somehow. For example, the vertically integrated diabatic heating could be converted to precipitation units and added to the precipitation anomalies or vice-

versa, the precipitation could be converted to a vertically integrated diabatic heating rate.

Typo's/wording:

l60: "extratropic"  "extratropical"

l83-84: "midlatitudes approximately doubles"  "midlatitudes is approximately double"

l138: "bonding"  "connected"

l228: "baroclinic property"  "baroclinicity"

l271: here and almost all subsequent occurrences of the word: "diabetic"  "diabatic"

l348: "search 40-year long simulations that exhibits"  "search for 40-year long periods within simulations that exhibit"

l648: "Differnces"  "Differences"

We would like to extend our appreciation to all reviewers for their time to assess our work, their positive feedback and constructive comments on our manuscript. These comments and suggestions have helped to improve the presentation and clarity of our main points. We have incorporated the reviewers' suggestions into the revised manuscript, mainly including:

1. Further analysis is conducted to demonstrate the role of the Rossby Wave Source (RWS, added new supplementary Fig. 9) in linking tropical forcing with extratropical responses, in order to address one reviewer's concern regarding a need to clearly show the physical mechanism underlying the tropical influence on waviness of the Northern Hemispheric mid-latitude circulation.

2. A more thorough assessment of relationships between tropical Pacific SST anomalies and midlatitude circulation changes is provided on the basis of AMIP-type simulations and Pacific pacemaker experiments. Since these AMIP-type runs cannot well simulate observed local SST-rainfall connections in the tropical Pacific, we don't present this additional analysis in the manuscript. This limitation also indicates that our heating-imposed experiment is necessary to directly examine the model's response to specified diabatic heating imitating the observed rainfall change in the tropics.

3. We put more emphasis on the discussion of the physical processes contributing to the enhanced waviness throughout the paper, rather than an understanding of whether these processes are internally or externally driven because some limitations of current models render an accurate attribution impossible, as suggested by Reviewer 2.

4. As suggested by the reviewers, we have moved original supplementary Fig. 4 back to the main text as new Fig. 4 to improve the readability.

5. Clean up a number of typos and wording issues raised by the reviewers.

Below are our detailed replies (in blue font) to each of their comments. We hope that we have sufficiently addressed all raised issues in the revised manuscript.

Reviewer #1 (Remarks to the Author):

The paper explores possible reasons of wavier jet in the Northern Hemisphere, that has been observed in the recent decades. Model simulations show that under modified tropical conditions in the Pacific + Maritime continent, wavier jet is found in a few simulations. The authors conclude that this forcing may be responsible for the NH wavier jet, rather than the Arctic Amplification, that had been suggested as the key reason by some previous studies.

I like the clarity of the manuscript in setting-up the research question. I have no

doubts that it falls within the scope of Nature Communications. However, there are a few major comments that need to be addressed before the manuscript can be considered for publication:

We thank the reviewer for the positive comments and constructive suggestions. Overall, we agree with the comments and have revised our manuscript to address them accordingly.

1. In the introduction, the authors state that there are two 'schools of thinking' on whether the wavier jet is caused by the Arctic Amplification (AA) or the other way round, i.e., wavier jet causes a warmer Arctic. While there are papers that, indeed, make a statement that the warmer Arctic is responsible for the wavier jet, the other statement remains a speculation, not a finding (Blackport and Screen (2020) explicitly say that they 'speculate'). While I can see how wavier jet can induce short-term temperature anomalies in the polar regions, I don't think anyone has been able to show that this is the mechanism of the recent trend in the Arctic.

Thanks for pointing this out. We revised that sentence as "This hypothesis has undergone rigorous scientific scrutiny since some studies showed contrasting results and speculated that the increased waviness is the cause of the meridional temperature gradient changes and not the other way around." in the Introduction (in lines 61 to 64) to make this statement more reflective of the state of science.

Furthermore, this manuscript shows that simulations with waiver jet have a cooler Arctic, not warmer. Hence, I believe that it is plausible to link a wavier jet to a tropical forcing, but connection to the AA is not supported by findings of this paper.

Thank you for this remark. This finding of the tropical connection is the novel part of our study, which as you indicated has not been specifically addressed in previous studies. Our comprehensive analysis of observations and various numerical simulations clearly elucidate that the wavy circulation structure along the westerly jet is closely associated with tropical forcing, regardless of the presence of anthropogenic forcing and AA in climate models.

2. The mechanism through which suppressed ITCZ convection and enhanced convection over the Maritime Continent trigger the formation of the CGT-like short-waves pattern along the jet stream suggested in the paper should be better supported by the results.

Firstly, no figure in the main manuscript shows any wave characteristics. Fig. 10 in the supplemental information shows the wave activity flux along with divergent wind component and velocity potential. This mechanism explaining the importance of the tropical forcing is a key finding and, hence, I believe that

Sup. fig. 10 should be moved into the paper.

But more importantly, I do not see how Sup. fig. 10 supports the mechanism offered in Fig. 6: there is no wave activity flux coming out of the Pacific or Maritime Continent (marked as 'wave source' region in Fig.6), neither in the observational nor in modelled data. I would like to see a plot that actually supports the proposed mechanism.

Thanks for letting us know this issue. We agree that we can improve the illustration of this linkage. To help readers see more clearly the importance of tropical forcing in forming the wave train pattern, we calculated the Rossby wave source (RWS) along with Velocity Potential (VP) and the wave activity fluxes across observations and all model simulations. Due to the space limitations, we present this new addition as Supplementary Fig. 9 and add relevant discussion in lines 278 to 283 as "first, the recent SST and precipitation trends in the tropical Pacific and its associated upper-level convergence/divergence anomalies (Supplementary Fig. 9) generate strong Rossby wave source (RWS, Methods) over the Western North Pacific, which in turn creates propagating Rossby waves at upper levels and subsequently disturbs the mean flow (Fig. 7a). This mechanism is supported by the observed and simulated RWS in Supplementary Fig. 9.", as well as a revision of the schematic diagram in Fig. 7(a).

Sardeshmukh and Hoskins (1988) have shown that upper tropospheric divergence/convergence in the tropics can interact with the mean flow to form the significant RWS in the subtropical westerlies which is able to effectively excite extratropical wave trains. Synoptically, the divergent circulation reaching subtropics can enhance the jet stream while amplifying the quasi-stationary waves through geostrophic adjustment and the potential vorticity dynamics. Since the wave activity fluxes are built on the basis of the QG approximation, this tool is more efficient to reflect group velocity of wave trains along the jet but less efficient to detect tropical related origins of these wave trains. Thus, a heating source embedded in the tropical easterly in the deep tropics, can still generate significant extratropical responses in the midlatitude westerly because of the RWS.

In observations, associated with tropical SST and rainfall changes over the past decades, there is clear Walker-like VP pattern in the upper troposphere displaying strong divergence over the Warm Pool Region and significant convergence over the tropical Eastern Pacific (TEP). The divergent circulation trend located to the north of the Warm Pool favors an increasing trend of the RWS over the Western North Pacific (WNP, denoted by the black box in Fig. R1), suggesting that tropical forcing, measured by the RWS in this area has gained intensity over the past decades, compared with other tropical regions along the same latitude. Thus, it is reasonable to expect that extratropical circulation is also sensitive to tropical forcing over the WNP on decadal time scales.

We apply the same calculation using the PI control and imposed heating experiments, in which enhanced waviness is successfully captured (Fig. R1). All these runs show that the generation of the CGT-like wave pattern along the jet is closely related to the generation of the RWS over the WNP. This serves as evidence that tropical forcing, both in observations and in our model, can excite the strong RWS over the WNP, which plays a key role to bridge the effect of tropical forcing to the downstream midlatitude region. This feature is supported by the wave flux activity patterns to some extent. Both in observations and the experiments, most of these fluxes along the jet display strong equatorward energy transport from the interior of Eurasia to East Asia and then appear to be reenergized once these fluxes pass the longitudes of the WNP. We hope this additional analysis will improve our reasoning regarding the tropical-midlatitude linkage and strength the argument that tropical forcing is important to enhanced midlatitude waviness along the jet.

Response Fig. 1. (a, c, e and g) The JJA Rossby wave source RWS and (b, d, f and h) velocity potential (VP), divergent wind component (black vectors) and wave activity fluxes (in the Northern Hemisphere; blue vectors) at 200hPa. The figures show results from: (a, b) JJA linear trends of ERA5 over 1979-2018; (c, d) simulated response in the sensitivity experiments; (e, f) regression between the Z200 SCI and RWS/ VP in 1800 years of CESM-LE PI simulations; (g, h) differences of RWS/ VP between the selected 23 positive and 16 negative cases in the pseudo-ensemble of CESM-LE PI runs, respectively. The RWS units are (a) $10^{-11} \text{ s}^{-2}/\text{decade}$ and (c, e, g) 10^{-11} s^{-2} . The VP units are (b) $10^5 \text{ m}^2/\text{s}/\text{decade}$ and (d, f, h) $10^5 \text{ m}^2/\text{s}$. The divergent wind component units are (b) $\text{m}/\text{s}/\text{decade}$ and (d, f, h) m/s . The wave activity fluxes units are (b) $\text{m}^2/\text{s}^2/\text{decade}$ and (d, f, h) m^2/s^2 , respectively. The Fig. R1 is also shown as Supplementary Fig. 9.

3. Lastly, a more detailed description of methods is needed (in lines 379-386). It is not clear to me how the correlation between fields was evaluated. What happened after the EOF analysis? What was the area that was compared? Did you estimate correlation between the PC2 (as PC2 is shown in Supplements, but it was not mentioned in the main manuscript) of the observed and modelled V200, their low-frequency components or something else? What are the two variables in eq. at line 383?

We mainly showed our EOF analysis in the supplementary materials due to the space limitations. As suggested by the reviewers, we added more detailed discussion of this part in the revision in lines 395 to 401.

The main goal of the EOF analysis is to put the recent 40 years observational change (Fig.1b) in the context of long term variability related to the CGT pattern, which represents a prevailing internal mode since 1850. The EOF2 of meridional winds at 200hPa in 3 reanalyses consistently capture a CGT-like pattern (spatial correlation within $20^\circ\text{N}-90^\circ\text{N}$, $0-360^\circ\text{E}$: ERA20C-EOF2 and OBS-trend (Fig.1b): $r = 0.56$;

ERA5-EOF2 and OBS-trend (Fig.1b): $r = 0.70$; NOAA20CR-EOF2 and OBS-trend (Fig.1b): $r = 0.52$) and exhibits strong interdecadal variations. In particular, in the past 40 years, a strengthening of this mode is evident, indicating that the enhanced waviness may in part represent a fluctuation of extratropical climate system. All follow-up analyses in the manuscript (e.g., the power spectrum and correlation analysis) are applied on PC2 and observed SST and precipitation.

We tested the statistical significance of the correlation coefficient in our manuscript via first calculating the lag-1 autocorrelation of each variable respectively (r_1 , and r_2) and then calculating the corresponding "effective sample size" (N^*) following equation 2. For example, when examining the significance of correlations between V200 PC2 and global SST from ERSST5, r_1 in eq. 2 is the lag-one autocorrelation coefficient of PC2; r_2 is the lag-one autocorrelation of SST in each grid. By plugging these two numbers into eq. 2, we can calculate "effective sample size" in each grid that will be eventually used to determine the threshold (based on critical values of the Student's T distribution) of significance. We have provided more details of this step in the corresponding part in the Method section of the main text.

Other comments:

I.98: This drying trend at 5N has been linked to a northward displacement of the ITCZ due to stronger warming of the NH, see, e.g.:
Schneider, T., Bischoff, T. & Haug, G. Migrations and dynamics of the intertropical convergence zone. *Nature* 513, 45–53 (2014). <https://doi.org/10.1038/nature13636>

High-latitude eruptions impacts on ENSO and AMOC
Francesco S. R. Pausat, Leon Chafik, Rodrigo Caballero, David S. Battisti
Proceedings of the National Academy of Sciences Nov 2015, 112 (45)
13784-13788; DOI: 10.1073/pnas.1509153112

Thank you for directing us to these references, which are now included in the revised manuscript.

I.186: I don't understand this statement. Recent Arctic SIC decline would be not captured if SIC variability below 30 years is filtered out.

We meant the Spatial Correlation Index (SCI; Fig. R2a), not a sea ice index. We construct it by calculating the spatial correlation between the JJA non-zonal Z200 trend from 1979 to 2018 in ERA5 (Fig. 1b) and the JJA non-zonal Z200 in each summer of the 1800-year CESM PI simulation within 20°N-60°N, 0°-360°. This index is used to measure how the PI control run simulates the CGT pattern on year-to-year time scales.

Response Fig. 2. (a) Z200 spatial correlation index (SCI) calculated as the spatial correlation between JJA non-zonal Z200 trend in ERA5 (from 1979-2018) and each of the 40-yr long JJA non-zonal Z200 trend derived from the CESM-LE PI 1800 year-long simulation (20°N-60°N). (b) Correlation between global SST and the Z200 SCI in the CESM-LE PI simulation. (c) Fourier power spectrum of the Z200 SCI (with the cycles longer than 10 years) from (a). The red, blue and green lines in (c) represent 95%, 90% “red noise” confidence and the “red noise” curve. The orange box highlights spectral peaks between 30-50yr. (d) Correlation between the 30-50 years band pass-filtered signals (that passes frequencies within a certain range ($1/50 \text{ year}^{-1} \sim 1/30 \text{ year}^{-1}$)) of the Z200 SCI and global SST in CESM-LE PI simulation. The dots in (b) and (d) indicate significant correlation above the 95% confidence level. The Fig. R2a, b and c are also shown as Supplementary Fig. 3.

I. 193: why Sup. Fig. 4 is not in the manuscript? This section refers to it a lot (at least five times), it would be easier to read if the figure was added to the paper.

Agreed. We have moved this figure to the main text and now it is new figure 4. Relevant discussion (lines 184 to 193) is added to improve the presentation of this part.

I.198: Is it global/hemispheric trend in Z200 /SST or in selected regions?

We make the composite based on the spatial correlation between non-zonal Z200 trend in ERA5 and the simulated counterpart derived from historical simulation of CESM-LE (40-member) within 20°N-60°N, 0 -360. We have made this point clearer in the revision in lines 186 to 188.

I.204: See my major Comment 1. Were you able to find any simulations were wavier jet was associated with a cooler tropical Eastern Pacific and a warmer Arctic at the same time?

We appreciate that the reviewer raised this concern. We did examine all members in the pseudo-ensemble that resembles the observed non-zonal Z200 trend over the past 40 years. There are 23 events satisfying our selective criteria, which is to show a high spatial correlation between observed non-zonal Z200 trend (1979-2018) and the simulated one within 20°N-60°N, 0°-360°. We look into these 23 events to further see how each member simulates tropical and the Arctic temperature variability separately. To better present this, we display the distribution of these 23 events in the scatter plot (Fig. R3) using the Pan-Arctic average surface temperature (TS) and tropical Eastern Pacific (TEP) average TS as the two coordinates. As expected, 70% of the 23 cases show a TS cooling signal over the TEP (17 cases cooling and 6 cases warming), while the Pan-Arctic surface temperature trends slightly prefer a cooling tendency (14 cases cooling and 9 cases warming). This indicates that when the model captures observed features of the CGT pattern (i.e., the model is constrained by the observed CGT pattern), it rather favors concomitant cooling in the tropics than warming in the Arctic. This statistical analysis indicates that 30% (8/23) of the 23 events, which successfully capture the CGT like Z200 trend in the midlatitudes, is associated with a cooler tropical Eastern Pacific and a warmer Arctic at the same time.

Response Fig. 3. (a) The composite map of surface temperature trends (TS, unit: K/decade) derived from those cases when the pseudo-ensemble (each consecutive 40-yr period) of the CESM-PI run shows the highest spatial correlation (20°N-60°N) with the observed non-zonal Z200 trend (defined as ‘23 positive cases’). (b) The scatter plot between domain-averaged TS in the tropical Eastern Pacific (20°S-10°N, 80°W-130°W) and in the Arctic region (70°N-90°N, 0°-360°) of the selected 23 positive cases.

I.299: can this wavelength be calculated?

Yes. The wave train exhibits a very clear zonal wave number 5 structure along the jet since the basic state around these latitudes favors the maintenance and propagation of zonal wavenumber 5 Rossby wave along the jet in summer (Ding and Wang, 2005). This feature is better presented below if we average the observed non-zonal Z200 trend pattern between 20°N-60°N in each longitude. Successive positive and negative

cells are emergent along this band with a clear zonal wave number 5 structure and rather even spacing. The wavelength can be correspondingly calculated as $360^\circ/5=72^\circ \sim 7200$ km.

Response Fig. 4. (a) Linear trend of JJA non-zonal Z200 (unit: m/decade) from 1979 to 2018 in ERA5. (b) Linear trend of JJA non-zonal Z200 averaged between 20–60° N in (a). The dotted areas indicate significant trends above the 95% confidence level.

Minor:

I.89: almost everywhere in the NH, but not the SH, where the trend pattern is more complex.

Rewritten, as suggested.

I.149: the long term trend over the past 40 years, if present, is ...

We have reworded accordingly.

I.168: Why 'subtle'? perhaps, replace with 'some'

We have reworded.

I.242: awkward wording

We have rewritten the sentence to make it clear that CO₂ forcing alone cannot explain the observed long term change of topical-Arctic temperature gradient. The new sentence is rewritten as: "In response to increasing anthropogenic forcing over the past four decades, the ensemble means of CESM-LE, CMIP5 and CMIP6 produce stronger tropospheric warming trends and enhanced precipitation over the TWCP than in observations (Fig. 5d, Supplementary Fig. 6e-f), which makes the long-term change

of the tropospheric tropical-Arctic temperature gradient in models behave differently from that in reality.”

l. 271, 301, ... : fix ‘diabetic ‘ heating (I saw it 4-5 times throughout the paper)

We have thoroughly corrected this sort of typos in the revision.

References:

Ding, Q. & Wang, B. Circumglobal Teleconnection in the Northern Hemisphere Summer. *J. Clim.* 18, 3483–3505 (2005).

Prashant D. Sardeshmukh and Brian J. Hoskins. The Generation of Global Rotational Flow by Steady Idealized Tropical Divergence. *J. Atmospheric Sci.* **45**, 1228–1251 (1988).

Reviewer #2 (Remarks to the Author):

Enhanced jet stream waviness induced by suppressed tropical Pacific convection during boreal summer

This paper examines the potential role of suppressed convection in the western tropical Pacific in driving trends in the extratropical circulation over recent decades. Previous studies have argued for the potential role of Arctic amplification in these "waviness" trends and this study puts forth an alternative hypothesis and argues that internally generated SST trends may have been responsible through their influence on tropical convection, wave generation and downstream teleconnections through the circumglobal waveguide. The arguments have been backed up by analysis of observed trends, internal variability in model simulations and in idealized experiments in which heating perturbations are imposed in the tropics within a GCM. Overall, I think this study is interesting and relevant and should be worthy of publication. However, I do have a number of comments that I recommend be considered before publication. While I don't see any of these as being particularly major, I think I have a sufficient number of them that, taken together, they amount to major revisions.

We appreciate the Reviewer's positive and constructive comments on our work, in particular the one directing us to available AMIP-type simulations and pacemaker experiments. We have examined those runs and discussed the insights gained from analyzing these experiments below. We have revised our manuscript to fully address the reviewer's concerns, as clarified in greater detail below.

General comments:

(1) Tightening up the arguments for whether or not the SST trend is internally generated: Currently, I think the authors may have been too quick to come to the conclusion that the observed SST trend is internal. This is stated throughout the text, and I don't think it has been adequately acknowledged that there is actually considerable debate over whether the cooling trend in the tropical Pacific may actually be anthropogenically forced and the models are doing something wrong (Seager et al 2019, Nature Climate Change, doi:10.1038/s41558-019-0505-x). I think that, given this debate, it's currently not ok to presume that the SST trends are internally generated. I think a valuable addition, would be to show whether the same magnitude of tropical Pacific SST cooling trends can be found in any members of the piControl pseudo ensemble and the historical ensemble. I recommend that the authors show that the real world tropical Pacific SST trend is not outside of the distribution from the model ensembles, before making the argument that it is

internal variability. But even then, I think it should still be acknowledged that there is debate over whether the SST trends are forced or internal.

We agree with the reviewer that it is premature to conclude that the observed SST trend over the tropical Eastern Pacific (TEP) since 1979 is mostly driven by internal variability considering that competing theories exist in explaining how tropical Pacific SST should respond to anthropogenic forcing and how the discrepancy between model simulations and observations are reconciled. In particular, current climate models possess some uncertainties, biases, and limitations to fully capture some fundamental atmospheric and ocean physics over the equatorial Eastern Pacific (Seager et al 2019; Wengel et al. 2021), which hamper us from establishing an attribution of the cause of the observed SST trend there with confidence. We have revised the text in lines 298 to 303 as “The next important question concerns the origin of this SST cooling over the TEP, yet current climate models lack the critical performance to establish a reliable attribution analysis. Considering that the enhanced waviness occurring in tandem with a tropical SST cooling can be detected in the PI simulation, we believe that the enhanced midlatitude waviness may be of internal origin, though we cannot rule out the possibility that anthropogenic forcing may also enhance the waviness through generating a SST cooling trend in the TEP.”

As suggested by the reviewer, we also examined how the CESM1-LE PI control and CESM-LE/CMIP5/CMIP6 historical runs simulate observed SST trends (non-global SST) in each longitude in the deep tropics (10°S-10°N). It is clear that neither the CMIP5/6 multi-model nor the CESM1 single-model ensemble means (the modeled forced responses to anthropogenic forcing) capture the observed trend over the TEP, as is the case for most of the individual CMIP5/6 models or CESM1 ensemble members. However their respective ensemble members encompass the observed trend within or around 1 standard deviation over other regions outside the TEP (Fig. R5 (b)-(d)). The deviation of observed SST cooling trend over the TEP from most of its CESM1-LE PI simulated counterparts is also apparent, which suggests that observed SST cooling over the TEP is hard to capture by current models. Nonetheless, we agree that it is still debatable on the causes of this discrepancy.

To further address these model limitations, in the revision we have rewritten several parts to highlight how tropical forcing can cause enhanced waviness in the midlatitudes, rather than whether the cooling is of internal or external origin. We also emphasize at the end of the paper (in lines 300 to 302) that the mechanisms underlying the tropical SST response to anthropogenic forcing remains an open question.

Response Fig. 5. Linear trends of JJA non-global sea surface temperature (SST, averaged between 10° S–10° N; unit: K/decade) from 1979 to 2018 in (a) the 1761 members of the CESM PI pseudo-ensemble, (b) the CESM-LE historical simulations; (c) CMIP5 historical+RCP8.5 experiments; (d) CMIP6 historical experiments (from 1979 to 2014), together with the non-global SST from ERSSTv5 analyses (red line). The global mean (60°S–60°N) is removed before calculations. The grey lines represent each of the individual model/member simulations and black lines indicate the ensemble mean. The green and blue lines represent 1 and 2 standard deviations of each ensembles, respectively.

(2) Being careful about statements regarding whether models are wrong or not and whether they can capture the observations. In the paragraph at l101 the observations are compared with the ensemble mean of the models and I think it could be better articulated what we get from this. I find the text to be a bit muddled as to whether it's saying the models are wrong or whether it's saying that the forced response does not resemble the real world trends. I think that latter is the conclusion that should be drawn but the use of statements like "CMIP6 exhibits some improvement" (l110) confuses matters because it makes it sound like we should be expecting the models to look like the real world. Then in the subsequent paragraph it refers to "These biases lead to the incorrect simulation...". This is very confusing. What biases? And if the claim that is made in the abstract is that the tropical Pacific SST trends are internal variability then we should not expect to see it in the CMIP ensemble mean which averages out the internal variability and shows the model simulation of the forced trend. I think this whole discussion needs to be improved. If you want to make arguments that what has been seen in the real world is consistent with the models and internal variability, then it needs to be shown that the ensemble spread encompasses the real world. Likewise, if you want to argue that the models are biased, it needs to be shown that the real world lies outside of the ensemble spread. I realize this has been done for the Z200 trends, but it hasn't been for everything else.

To complement our previous calculations regarding the comparison of the observed Z200 trends in the ensemble mean and in each individual member, we show that the

observed SST cooling and ITCZ drying over the tropical Pacific also lie out of the 2 standard deviations of the CESM-LE spread, or the CMIP5 and CMIP6 models (Fig. R6).

However, as the reviewer suggested, due to models' bias in reflecting the tropical SST response to anthropogenic forcing over the TEP, it would be premature to determine the causes of the difference between the observation and the ensemble mean of current climate models. Thus, we have rewritten several parts of the paper in lines 304 to 306 to emphasize that the main goal of this work is to provide a new perspective to explain an alternative physical process that is partially responsible for the enhanced waviness in the midlatitudes observed in the recent past.

Response Fig. 6. Differences of JJA linear trends in (a, c and e) non-global SST (calculated as the SST pattern with the global mean SST removed in each grid; unit: K/decade) and (b, d and f) precipitation (unit: mm/day/decade) between the observations and ensemble means of (a-b) CESM-LE historical simulations, (c-d) CMIP5 historical+RCP8.5 experiments, (e-f) CMIP6 historical experiments, respectively. Hatching indicate areas where the observed trends lie outside 2 standard deviations of ensemble means of the historical runs.

(3) I'd recommend some improved discussion of the relative magnitudes of aspects discussed in the figures. Wildly different color bars are used for the obs and the models but from eyeballing it, it does look like the magnitude of e.g., the SST anomalies relative to the Z200 anomalies is quite comparable. I

recommend you do what you can to try to make the color bars similar to one another, but I understand that may not be possible, so I'd recommend at least adding some discussion about the relative magnitudes and whether they match up or not.

Thanks for your suggestion. We have added more discussions to clarify the different magnitudes between observations and the historical model simulations in the revision (lines 109 to 111).

We primarily compare observations with the ensemble means of various models that exhibit smaller magnitude or opposite sign patterns to those observed. If we use the same color scale for the same variable in both observations and models, readers may have a false impression that models are way worse to replicate some observed features. To avoid this problem, we still use different colors to highlight some detailed structures of models' response. For example, the non-zonal trend of Z200 and V200 (between $-4 \sim 4 \text{ m}\cdot\text{decade}^{-1}$ and $-0.3 \sim 0.3 \text{ m}\cdot\text{s}^{-1}\cdot\text{decade}^{-1}$) in the model are weaker than the observed values ($\pm 12 \text{ m}\cdot\text{decade}^{-1}$ and $\pm 1 \text{ m}\cdot\text{s}^{-1}\cdot\text{decade}^{-1}$).

(4) An obvious analysis to perform that would back up the arguments made in the paper is to consider AMIP-type simulations or pacemaker simulations and I'm not sure why this hasn't been done. A suite of relevant experiments accompany the CESM1 large ensemble http://www.cesm.ucar.edu/working_groups/CVC/simulations/cam5-prescribed_sst.html. They may stop short of the exact years that are used in this analysis, but there are equivalent simulations available with CESM2 as well, that extend to present http://www.cesm.ucar.edu/working_groups/CVC/simulations/cam6-prescribed_sst.html. If the authors arguments are correct and the tropical Pacific trends are a primary influence on the circulation trends, then shouldn't we expect to see these trends in prescribed SST simulations? A comparison between the coupled runs and either GOGA or TOGA simulations seems like an obvious comparison to make and I'm wondering why the authors haven't done this.

Thanks for your suggestion. We were aware of the availability of these runs when we started to conduct this analysis. The main reason that we had not included these runs in our study was that those experiments are forced by both specified SST (or nudged SST) and anthropogenic forcing, which makes our interpretation of the results difficult since we definitely need a control version of the same run in which climatological SST should be specified in the tropical domain in the TOGA or pacemaker runs, while anthropogenic forcing is still allowed to vary. Unfortunately, these corresponding control simulations are not yet available so we cannot isolate the response of the model to only the imposed tropical SST forcing.

However, here we discuss these results for the information of the reviewer. In both the AMIP type TOGA runs using CESM1 and CESM2 (Fig. R7 and Fig. R8), the observed tropical rainfall trend in the Pacific is still not well captured with models simulating increasing rainfall over almost everywhere in the Warm Pool. This pattern is in stark contrast with the observed pattern because in the real world, the formation of rainfall and SST anomalies over there is owing to an active two-way coupling process (Wang et al. 2005). However, in the AMIP run, this two-way interaction is artificially interrupted. Without a correct simulation of tropical rainfall pattern in these two TOGA runs over the WNP, it is not very informative to further examine its response in the extratropics.

We also check the pacemaker run (Fig. R9), in which the tropical SST is nudged to the observed SST over the Pacific domain and varying CO_2 forcing is imposed. Some discrepancies are still seen in the simulation of tropical rainfall over the WNP (an increased rainfall anomaly centered over 160°E) but its overall resemblance to the observed pattern is better than that in the TOGA runs. Thus, the extratropical responses are also improved. However, since CO_2 forcing is present in this pacemaker simulation and the effect of CO_2 forcing cannot be clearly removed, we still do not have an accurate depiction of the response to the nudged SST over the Pacific domain without external forcing. To overcome these limitations, we conducted our novel experiments by directly impose diabatic heating sources over the key regions to imitate the observed tropical rainfall variability while muting the anthropogenic forcing.

Response Fig. 7. Linear trends in JJA (a) Z200 (unit: m/decade) (b) non-zonal Z200 (unit: m/decade) (c) precipitation (unit: mm/day/decade) (d) SST (unit: K/decade) (e) “non-global SST” (calculated as the SST pattern with the global mean SST removed in each grid; unit: K/decade) from 1979 to 2017 (1979-2005: historical simulations, 2006-2017: RCP8.5 experiments) from CAM5 Prescribed SST AMIP 10-member ensemble means.

Response Fig. 8. Linear trends in JJA (a) Z200 (unit: m/decade) (b) non-zonal Z200 (unit: m/decade) (c) precipitation (unit: mm/day/decade) (d) SST (unit: K/decade) (e) “non-global SST” (calculated as the SST pattern with the global mean SST removed in each grid; unit: K/decade) from 1979 to 2014 from CAM6 Prescribed SST AMIP 10-member ensemble means.

Response Fig. 9. Linear trends in JJA (a) Z200 (unit: m/decade) (b) non-zonal Z200 (unit: m/decade) (c) precipitation (unit: mm/day/decade) (d) SST (unit: K/decade) (e) “non-global SST” (calculated as the SST pattern with the global mean SST removed in each grid; unit: K/decade) from 1979 to 2013 from CESM1 Pacific Pacemaker 20-member ensemble means.

(5) All the figures were kind of fuzzy in the pdf version I had and I think it's strange in Figs 1 and 2 to have panel (c) to the right of (d). Suggest reordering the panel labelling.

We will upload all figures in pdf format in this round of submission. We have also reorganized the panel labels in Figure 1 and Figure 2 (and supplementary Fig. 1 & 2).

Comments by line number:

I33: At the outset here in the abstract it may not be clear to readers what "tropical internal cooling" is. I realize you may be up against a word limit but I'd recommend "for the role of internally generated tropical cooling"

The abstract has been rewritten to emphasize that tropical Pacific SST cooling is important to drive the enhanced waviness. We have also removed "internal" and related words from the abstract since we feel it is more appropriate to focus on the physical process itself in this study instead of its internal origin.

I41: It's not clear what "jet persistency" means or how it really relates to the findings. Perhaps "jet waviness" would be more appropriate?

Thanks for the suggestion. We changed "jet persistency" to "jet waviness" to make the statement more accurate.

I43: I think references are needed for these studies that have "expressed concern about the increasing meandering". Perhaps it's some of the references mentioned in the following sentence, but I think it would be appropriate to cite the relevant ones here.

We have added references (Cvijanovic, I., et al., 2017; Cohen, J., et al., 2018; Coumou, D. et al., 2018; "A tug-of-war over the mid-latitudes.", 2019; Matsumura, S., et al., 2019) in the revised manuscript.

I47: "enhanced amplitudes" of what? stationary waves?

We have rewritten this sentence as "Prevalent heatwaves and flood-producing storms across the midlatitudes have been linked to enhanced amplitudes of atmospheric quasi-stationary wave trains along the jet stream." in the revised version of the manuscript.

I59: "AA amplifies midlatitude stationary waves via the reduction of the equator-to-pole thermal gradient". I'm not sure that this is anchored in theory discussed in reference 19. I haven't read through the whole of that paper before making this comment, but I did check the "Tropospheric dynamics" section and I think it's arguing the opposite. It's arguing that an increased equator-to-pole temperature gradient will increase the baroclinicity and, therefore, the growth of atmospheric eddies. What the authors are referring to may be addressed in another section of ref 19 that I didn't read carefully, but I suggest they double check. In any case, I think it should also be acknowledged that the theory behind the enhanced waviness in association with arctic

amplification is not at all clear cut, as discussed by Woollings and Hoskins (2015) doi:10.1007/s40641-015-0020-8. l59:

We have rewritten this part to put this statement more objectively and also in the context of more updated studies (Hoskins and Woolings, 2015) on the topic. We agree with the reviewer that our current understanding of this topic is still limited and the response of midlatitude circulation to anthropogenic forcing and associated changes in the Arctic and the meridional temperature gradient depends on the season, hemisphere, vertical level, and spatial and temporal scale. To address such complexity, other factors including baroclinicity, the strength of zonal winds, moisture changes, diabatic heating over the midlatitudes, effects of orography, a vertical asymmetry of temperature gradient changes, stratospheric processes and many more will have to be considered to understand this issue.

So, this part of the text has been rewritten as “The scientific community has not reached a consensus on the aforementioned two sources of increased jet stream waviness nor has acquired a complete understanding of the sensitivity of various properties (e.g., wavelength, amplitude and vertical structure) of midlatitude stationary waves to different climate drivers.” in lines 67 to 70.

l94: suggest "widespread cooling"  "widespread relative cooling" or "widespread cooling relative to the global average", since a cooling is not really apparent in Fig 1e.

We have modified "widespread cooling" to "widespread relative cooling" to make it clearer and more accurate.

l168: "subtle importance" is quite ambiguous. It sounds like you're arguing it is important but it's subtle. But what I think you're trying to say that it's not really important or maybe something like it "plays a smaller role compared to the relative cooling of the tropical Pacific"

Thanks for pointing this out. We rewrote this sentence as "This discrepancy suggests that the observed Arctic warming alongside the enhanced waviness in the midlatitudes is not a primary source for the generation of the CGT-like circulation trend in the simulations."

l172: This section gets a bit frustrating with the continued reference to the supplementary material. If I understand correctly, the difference between Fig 3 and supplementary Fig 3 is that Fig 3 is for 40y trends and Fig S3 is for interannual variability. It may be less frustrating for readers if Supplementary Fig 3 is not given so much of a write up and is instead mentioned in the paragraph above with a simple statement that similar patterns are found when considering interannual variability and with some mention of the power

spectrum. The same kind of goes for the paragraph at l188. It's described as a whole new analysis but we're only pointed toward supplementary figures, which gets frustrating, but really this is just confirmation that the same thing is seen in the historical simulations as in the piControl.

Thanks for the suggestion. We have shortened the discussion of supplementary Figure 3 and moved supplementary Figure 4 back to the main text as new Fig. 4 to improve the readability and presentation.

l217: It's not clear what "enhanced atmospheric variability" means here. I think instead of "enhanced atmospheric variability" you perhaps mean "atmospheric circulation trends"?

What we meant was atmospheric circulation trends. We have rewritten the sentence accordingly.

l280: I think it's an overstatement to say that the tropical forcing is "essential" to generating this wave train structure. Given the arguments related to barotropic instability of the jet exit region in the Atlantic, it seems likely that there are many different perturbations that could end up giving this structure if it is the way in which the mean flow is unstable e.g., Simmons et al (1983), JAS, 40, 6, 1363--1392). Suggest rewording.

We have rewritten that part to emphasize that tropical forcing plays a partial role to generate the wave train structure in lines 262 to 263 as "The strong low pressure center generated over the Northeast Atlantic suggests that the added tropical forcing enhances barotropic instability at the jet exit region". We also cite Simmons et al (1983) in the revision.

l327: I'm not sure that the reference to Fig 6c is correct here since that is the schematic.

This was a typo. We have corrected it.

l330: Fig 4c doesn't show SSTs, it shows atmospheric temperature.

This was also a typo. The Fig. 4c (now Fig. 5c) shows atmospheric temperature from ERA5 and the ensemble mean of CESM-LE 40-member historical runs. We have corrected it.

l358: Recommend being more specific about whether this is CESM1 or CESM2 and adding a reference.

It is CESM1. We have made this point clear in the revision and have added the reference.

I381: I appreciate the consideration of the temporal degrees of freedom, but I wonder if also the spatial degrees of freedom should be accounted for by considering the False Discovery Rate (Wilks (2014) "The stippling shows statistically significant grid points", BAMS)

Thanks for your suggestion. The most appealing finding in our study is that the composite result derived from the pseudo-ensemble indicates tropical SST cooling over the TEP is closely associated with the wavy pattern in the midlatitudes. However, it is a little bit hard to apply the FDR procedure directly to examine the influence of the spatial degrees of freedom in this regional feature since spatial autocorrelation of SST data over that area is sensitive to the land-sea configuration surrounding the TEP. Alternatively, we use a similar method as that proposed by Livezey and Chen (1983) to examine the sensitivity of our result to spatial degrees of freedom: "estimating the frequency distribution for numbers of locally significant tests using Monte Carlo methods (i.e., randomly resampling the available data in a manner consistent with the global null hypothesis").

The original composite plot (23 positive events minus 16 negative events; Fig. R10a) show that a significant SST cooling pattern over the TEP is closely associated with the formation of wave train pattern in the midlatitudes. To test that the clustering of many significant SST composite values over the TEP are meaningful and not only due to strong spatial autocorrelation of SST over that area, we redo the composite 1000 times using randomly selected 23 and 16 events from the entire pool. After this procedure, we can obtain 1000 new composite SST fields. In each grid, we can determine the threshold at the 95th significance level (a two-tailed problem) using the average of 25th +975th values if we sort 1000 composite values in an ascending order in that grid. The significance of our original composite is reevaluated using this set of new thresholds in the domain (Fig. R10b), which display a quite similar pattern as the original one examined by the two-sample t-test.

We further calculate how many grid points over the TEP are statistically significant by the two sample T-test in each randomly constructed composite (Fig. R10c). It seems that most of composite fields only contain less than 200 grids with significant values over the region. The pdf of these 1000 counts shows that the 95 percentile is 303 grids (the 99 percentile is 569 grids), which is much lower than what we have had in the original composite (733 grids). So, we are confident that the spatial autocorrelation of SST over TEP is not the main reason for the many significant composite SST values in the region, but the composite pattern seen in Fig. R10 has physical meaning. We also do the same test for the precipitation composite (Fig. R11) to find that the dipole-like rainfall pattern over the Warm pool has very little influence from this spatial-autocorrelation problem.

Response Fig. 10. (a) The differences of SST (unit: K/decade) between the selected 23 positive and 16 negative cases in the pseudo-ensemble of CESM PI runs. The cross-hatched areas denote the significant differences based on the two-sample t-test, $p < 0.05$. (b) The same as (a), but the hatching indicates statistical significance using a randomly resampling test (Monte Carlo) method. (c) The probability density function (PDF) for the number of P-values ($\alpha = 0.05$) over the tropical Eastern Pacific (TEP; 15° S- 15° N, 90° W- 180°) after randomly redoing composites (23 positive cases minus 16 negative cases) for 1000 times. The red box in (a) shows the SST domain over the TEP. The Fig. R10a is also shown as Fig. 3c.

Response Fig. 11. The same as response Fig. 10, but for the precipitation composite. The red box in (a) shows the PREC domain in the tropical West-Central Pacific (TWCP; 15° S- 15° N, 100° E- 150° W). The Fig. R11a is also shown as Fig. 3d.

Figure 4 c and d. I think it would be useful here to have stippling or shading to indicate where the reanalysis trends lie outside of the ensemble distribution of trends.

Thanks for your suggestion. We calculated the standard deviation of vertical TA and precipitation trends of CESM-LE 40-member historical runs and use 2 standard deviations as a criterion to determine the spreads of ensemble members. The shaded areas in Fig. R12c&d indicate that the observed tropospheric temperature trend lie outside 2 standard deviations away from the forced trend (the ensemble mean) in each grid in the tropics. The observed rainfall along the ITCZ and over the Maritime Continent are also very different from (>2 standard deviations) those forced trends by the model.

Response Fig. 12. JJA zonal mean component of air temperature (TA) trend profile from (a) ERA5 (1979-2018, unit: K/decade) and (b) the ensemble mean of CESM-LE 40-member historical runs (1979-2018, unit: K/decade), respectively. (c) Differences of zonal mean TA trends between ERA5 and the ensemble mean of CESM-LE 40-member historical runs. (d) Differences of JJA precipitation trends (60°S-60°N; unit: mm/day/decade) between ERA5 and the ensemble mean of CESM-LE. The shaded areas indicate the ERA5 trends lie outside 2 standard deviations away from the ensemble mean of CESM-LE 40-member historical runs. The Fig. R12 is also shown as Fig. 5.

Figure 4: For the discussion surrounding this figure, I think there needs to be some acknowledgement that the overall diabatic heating perturbation that the model is seeing is the combination of that due to the precipitation anomalies plus the heating that is being imposed. We don't have a good sense of what the magnitude of that imposed heating perturbation is. I recommend showing or discussing that somehow. For example, the vertically integrated diabatic heating could be converted to precipitation units and added to the precipitation anomalies or vice-versa, the precipitation could be converted to a vertically integrated diabatic heating rate.

We have added a statement in the revision in lines 265 - 268 as “Since these imposed heating will trigger complex responses of processes associated with cloud physics, vertical diffusion and radiation in the tropics, the overall diabatic heating to determine the extratropical response is a combination of imposed and induced diabatic heating anomalies in the model”. The goal of the imposed heating experiment is to examine how a diabatic heat source that imitating the observed rainfall trend in the tropics excite extratropical response in the model. To determine how much heating rate we need to add in the model, we calculate this rate based on a simple scaling between latent heat release and condensation: that is a 1mm/day rainfall anomaly is equivalent to a uniform 0.25K/day heating rate throughout the air column from the surface to the

top of atmosphere (air mass is about 1000hPa). However, in reality, the diabatic heating is released mostly in the middle of the troposphere and this heat is only used to warm the air column below the tropopause (200hPa; air mass is about 800hPa). Thus, the maximum heating rate in the mid-troposphere should be around 0.25×2 (reaching the maximum at 500hpa and decreasing to zero at the surface and 200hPa) $\times 1.25$ (adjustment due to the less mass below 200hPa) = 0.6. In addition, horizontally, the heating rate has the maximum at the center of the imposed domain and drop to zero at the edge, this heating rate at the center at the mid-troposphere should be further adjusted by $0.6 \times 2 = 1.2$.

Thus, a heating source in our idealized vertical and horizontal shape, resulting from a 1mm/day rainfall anomaly, should have the maximum heating rate of 1.2 K/day at the 500hpa at the center. In observations, the rainfall rate is about 0.4-0.5 along the ITCZ, thus we add ± 0.5 heating rate over the centers of the two places to approximately mimic diabatic heating rate possibly released by the rainfall anomalies. We describe this estimation more clearly in the revision.

Typo's/wording:

l60: "extratropic"  "extratropical"

Corrected.

l83-84: "midlatitudes approximately doubles"  "midlatitudes is approximately double"

Agree. Done.

l138: "bonding"  "connected"

Agree. Done.

l228: "baroclinic property"  "baroclinicity"

Corrected.

l271: here and almost all subsequent occurrences of the word: "diabetic"  "diabatic"

All these typos are corrected.

l348: "search 40-year long simulations that exhibits"  "search for 40-year long periods within simulations that exhibit"

Thanks for the correction.

l648: "Differnces"  "Differences"

Done.

References:

Coumou, D., Di Capua, G., Vavrus, S. *et al.* The influence of Arctic amplification on mid-latitude summer circulation. *Nat Commun* **9**, 2959 (2018). <https://doi.org/10.1038/s41467-018-05256-8>.

Lee, M. H., Lee, S., Song, H.-Y. & Ho, C.-H. The recent increase in the occurrence of a boreal summer teleconnection and its relationship with temperature extremes. *J. Clim.* **30**, 7493–7504 (2017).

Wang, S. Y., Davies, R. E. & Gillies, R. R. Identification of extreme precipitation threat across midlatitude regions based on short-wave circulations. *J. Geophys. Res. Atmos.* **118**, 11059–11074 (2013).

A tug-of-war over the mid-latitudes. *Nat Commun* **10**, 5578 (2019). <https://doi.org/10.1038/s41467-019-13714-0>.

Matsumura, S., Kosaka, Y. Arctic–Eurasian climate linkage induced by tropical ocean variability. *Nat Commun* **10**, 3441 (2019). <https://doi.org/10.1038/s41467-019-11359-7>.

Cohen, J., Pfeiffer, K. & Francis, J.A. Warm Arctic episodes linked with increased frequency of extreme winter weather in the United States. *Nat Commun* **9**, 869 (2018). <https://doi.org/10.1038/s41467-018-02992-9>.

Cvijanovic, I., Santer, B.D., Bonfils, C. *et al.* Future loss of Arctic sea-ice cover could drive a substantial decrease in California's rainfall. *Nat Commun* **8**, 1947 (2017). <https://doi.org/10.1038/s41467-017-01907-4>.

Yanai M, Tomita T. 1998: Seasonal and interannual variability of atmospheric heat sources and moisture sinks as determined from NCEP-NCAR reanalysis. *Journal of Climate*, **11**, 463-482, doi: 10.1175/1520-0442(1998)011<0463:SAIVOA>2.0.CO;2.

Wang, B., Ding, Q., Fu, X., *et al.* (2005) Fundamental challenge in simulation and prediction of summer monsoon rainfall, *Geophys. Res. Lett.*, **32**, L15711, doi: 10.1029/2005GL022734 .

Wengel, C., Lee, S.S., Stuecker, M.F. *et al.* Future high-resolution El Niño/Southern Oscillation dynamics. *Nat. Clim. Chang.* **11**, 758–765 (2021). <https://doi.org/10.1038/s41558-021-01132-4>

Petoukhov V, Rahmstorf S, Petri S, Schellnhuber HJ (2013) Quasiresonant amplification of planetary waves and recent Northern Hemisphere weather extremes. *Proc Natl Acad Sci USA* **110**(14):5336–5341.

Livezey, R.E. and Chen, W.Y. (1983) Statistical Field Significance and Its Determination by Monte Carlo Techniques. *Monthly Weather Review*, 111, 46-59.

REVIEWER COMMENTS

Reviewer #1 (Remarks to the Author):

The authors have done a thorough job in addressing the comments from the first review. The amount of new information in the new their response is overwhelming.

I particularly like Response Fig. 3b. I think the information shown in this plot supports the statement that there is 'less of an influence from the Arctic amplification on the observed mid-latitude 40 wave amplification than what was previously estimated'. I can suggest adding it to the manuscript. If I am interpreting it right, then only one ensemble member with amplified wave pattern in NH extratropics showed weak warming over the Arctic and weak warming over the Eastern Pacific. And there were five ensemble members that had amplified Z200 EOF2 associated with the warmer Eastern Pacific and colder Arctic. This suggests that the equator-to-pole gradient might be important for enhanced waviness. On the other hand, in case of the cooler Eastern Pacific, the odds of getting a wavier jet are about the same in case of a warmer and cooler Arctic.

Further to that, I am very interested in which of the two anomalies - increased rainfall over the Maritime continent or cooling of the Equatorial Pacific (or both) - play the key role. I understand that this east-west pattern may be robust and these two anomalies usually occur together. However, it would be good to separate them in the experiments by adding heating over the Maritime continent only or cooling over the Pacific only. These experiments may be helpful in separating the role of rainfall increase followed by divergent wind from the Maritime Continent (Response Fig. 1b) and increased equator-to-pole gradient (suggested by Fig.7 and Response Fig. 3b).

Depending on the results of separate heating/cooling experiments, Fig. 4e,f may use anomalies over the Maritime continent along X Axis rather than anomalies in the red box (this makes more sense from Response Fig. 1b anyway)

Fig. 1b and Resp Fig4 show that wave-like anomalies over the extratropical Pacific are displaced further north compared to the latitude of the wave pattern over the rest of the hemisphere. It reminds me the pattern in Fig. 1c in Choi and Kim, 2019. Did you think that there might be two interacting waves: one in the Pacific, triggered by anomaly over the Maritime continent, and a circumglobal wave in the NH mid-latitudes? This is something that can be suggested reading I. 288-290, but remains vague.

Further to that, reading the Caption to fig. 7, I think that a RWS in the Western Pacific by itself cannot enhance barotropic instability around the jet exit over the Northeast Atlantic. For me, it is a good argument to support the idea of two interacting waves. Otherwise, please explain better in the Caption.

In the response letter, Resp Fig 1b shows a large VP trend centred over the Maritime continent, that is made responsible for RWS trends further north. The problem I have with this argument is that I don't see similar RWS trends in the SH. Why is that?

Overall, I am happy with the revised manuscript, but I believe that the paper will benefit from additional experiments with separate heating anomalies.

Minor:

Fig. 5a: ERA5 has a problem with low stratospheric temperatures in 2000-2006. For this plot I suggest using ERA5.1 if that has not been used, though it may not be particularly important for a long term trend. (<https://confluence.ecmwf.int/pages/viewpage.action?pageId=181130838>)

I.303: the convection is increased over the Maritime continent, please be more accurate

L.402, 403: I suggest omitting 'at the same level'

I. 403: Coriolis -> planetary

I.403: is rotational component used in this paper?

References:

Choi, W., Kim, KY. Summertime variability of the western North Pacific subtropical high and its synoptic influences on the East Asian weather. *Sci Rep* 9, 7865 (2019).
<https://doi.org/10.1038/s41598-019-44414-w>

Reviewer #2 (Remarks to the Author):

Enhanced jet stream waviness induced by suppressed tropical Pacific convection during boreal summer, by Sun et al

I appreciate the authors efforts to address my comments from the previous round. I am mostly satisfied that they have adequately addressed them, but I would like to comment again on one of the aspects I commented on last time along with a few minor comments listed below. For my general comment I leave it to the authors whether they wish to address this (or to the editor to decide whether they feel this is necessary) as I don't see it as being a show-stopper to publication but just something I think would be beneficial to show.

General comment: I am glad the authors investigated the GOGA, TOGA and pacemaker results in the response to my review. I accept that the precipitation response in the prescribed SST experiments is not represented well enough, but I think this would be worthwhile information for readers to have, so I think it could be worth mentioning. But the main point that I'd like to make is that the authors said that they couldn't use the pacemakers to explore the impacts of the tropical Pacific SST and precip trends because they didn't have a control to compare with. But I don't think this is true. The fully coupled simulations of the CESM1 large ensemble can be used as the control to compare with. The only difference between the fully coupled CESM1 large ensemble and the pacemaker simulations is that the pacemaker simulations have the trend in the tropical Pacific SSTs imposed. It seems like if the wavetrain is more like the observed in the pacemaker simulations than it is in the LENS2 simulations then it would help the authors case. The pacemaker members could be added to Figure 4a for example. If they consistently show a higher correlation than the CESM-LE members then it would indicate that having the right SST trend in the eastern tropical Pacific helps push the model toward the observed trend. If they don't there's still the possibility that discrepancies in the precip trends might be messing things up. But my main point is that I don't think the lack of a control renders these simulations unusable for this purpose because the CESM large ensemble can be the control so if they help make the case for the importance of trends in the tropical eastern Pacific then perhaps it is worth adding that additional piece of evidence in.

I187: In the sentence before this it's stated that some CESM-LE members capture the observed changes better than others, even though the ensemble mean fails to replicate the observed wave pattern. Now here it is then stated that this "infers the role of the aforementioned tropical forcing in contributing to the spread of the simulated recent changes in the mid-latitude circulation". I don't think this is what infers that - it's after the subsequent discussion where this can then be inferred. At this point, I think the only thing that can be inferred from the fact that some members capture the pattern better than others even though the ensemble mean doesn't totally produce it is that internal variability may play an important role.

I100: Maybe point to Fig 1g here when discussing the precipitation changes?

I64: "role in the change"  "role of the change"

I68: "nor has aquired"  "nor has it aquired"

I192: For consistency with the way the latitudes are presented, perhaps you should put degrees symbols on the longitude range.

I246: "which resemble the CGT"  "in a manner that resembles the CGT"

Supplemental Fig 6: typo in caption "differnces"  "differences"

We would like to thank the two reviewers for the time they have devoted in reviewing the previous versions of the paper and for their constructive comments. These comments and suggestions continuously helped improve the presentation and clarity of this paper. The main changes we incorporated in this revision include: 1) conducting new experiments aimed at investigating the relative roles of ITCZ drying and more rainfall anomalies over the Maritime Continent (MC) in causing waviness along the jet; 2) adding more discussion of the comparison between the Pacemaker and CESM-LE experiments; 3) examining the temperature trend by combining ERA5 and ERA5.1 over the common period. We also cleaned up all minor problems pointed out by the reviewers. Below are our detailed replies to each of their comments. We hope that we have sufficiently addressed all comments in the revised manuscript. In the responses, words in black are the original reviews, followed by our replies in blue.

Reviewer #1 (Remarks to the Author):

The authors have done a thorough job in addressed the comments from the first review. The amount of new information in the new their response is overwhelming.

We appreciate your positive comments on our previous revision. We hope that in this new revision we have fully address all the remaining issues.

I particularly like Response Fig. 3b. I think the information shown in this plot supports the statement that there is 'less of an influence from the Arctic amplification on the observed mid-latitude 40 wave amplification than what was previously estimated'. I can suggest adding it to the manuscript. If I am interpreting it right, then only one ensemble member with amplified wave pattern in NH extratropics showed weak warming over the Arctic and weak warming over the Eastern Pacific. And there were five ensemble members that had amplified Z200 EOF2 associated with the warmer Eastern Pacific and colder Arctic. This suggests that the equator-to-pole gradient might be important for enchanted waviness. On the other hand, in case of the cooler Eastern Pacific, the odds of getting a wavier jet are about the same in case of a warmer and cooler Arctic.

Thanks for this suggestion. Your interpretation is in line with ours. We have added Response Fig. 3 from our previous response as Supplementary Fig. 3 in this new revision, and have added relevant discussions as "Of these 23 cases, 9 cases (40%) show a slight warming trend in the Arctic and 17 cases (74%) exhibits a tropical SST cooling over the TEP (Supplementary Fig. 3)." in lines 164 to 166 in the manuscript.

Further to that, I am very interested in which of the two anomalies - increased rainfall over the Maritime continent or cooling of the Equatorial Pacific (or both) - play the key role. I understand that this east-west pattern may be robust and

these two anomalies usually occur together. However, it would be good to separate them in the experiments by adding heating over the Maritime continent only or cooling over the Pacific only. These experiments may be helpful in separating the role of rainfall increase followed by divergent wind from the Maritime Continent (Response Fig. 1b) and increased equator-to-pole gradient (suggested by Fig.7 and Response Fig. 3b).

We appreciate that you raise this point. According to your suggestion, we performed two additional experiments following the setup of the original one to separately examine the response of the model to MC heating and ITCZ cooling (Response Fig. 1). What we learned from these two new experiments alongside the original one, is that the ITCZ cooling is more essential than the MC heating in contributing to the formation of the wave train structure along the jet. In the ITCZ cooling experiment, the tropical rainfall response can well capture an elongated drying zone along the ITCZ and an enhanced precipitation over the MC. In contrast, the MC heating experiment cannot produce either suppressed precipitation along the ITCZ (Response Fig. 1k, l) or enhanced rainfall over the MC, indicating the importance of ITCZ rainfall anomalies in governing the entire tropical rainfall pattern in the deep tropical Pacific.

The Z200 response from both experiments show amplified wave trains in the NH midlatitudes, but the circumglobal feature is clearer in the ITCZ cooling run, while the MC heating experiment seems to generate a displaced pattern with respect to the jet, especially over the North Pacific. In both cases, a low-pressure center at the jet exit region over the Northeast Atlantic was created, indicating an essential role of tropical forcings – no matter where they are located – in triggering strong barotropic instability over the jet exit region.

The spatial correlation of Z200 response in the ITCZ cooling, MC heating and original experiments with the observed trend pattern is 0.52, 0.01 and 0.68, respectively within the extratropics (20°N - 60°N , 0° - 360°), further reinforcing that the ITCZ cooling dominates the tropical-CGT connection revealed in our original heating-imposed experiment. We have briefly discussed this finding in the method part of the revised manuscript, which highlights the active role of ITCZ variation in determining the large-scale rainfall pattern in the tropical Pacific. One reason behind this active role is probably the importance of ITCZ-related convection anomalies in regulating the structure, locations and intensity of the Walker Circulation in the tropics.

Response Fig. 1. (a, b, c, d) The Z200 (unit: m), (e, f, g, h) non-zonal Z200 (unit: m) and (i, j, k, l) precipitation (unit: mm/day) responses in the CESM1 to additional heating sources added over the Maritime Continent (MC) and (or) the ITCZ. The green (10°S-0°, 110°E-150°E) and brown (0°-10°N, 120°E-80°W) colour filled ovals denote the locations where a pair of positive and negative heating sources are added in the CESM1. The figures on the first, the second, the third and the fourth columns are results from ERA5, two-heating experiments (the original set), ITCZ cooling- only and MC heating-only experiments, respectively. The cross-hatched areas denote the significant differences between the CTL and imposed-heating experiments based on the two-sample t-test, $p < 0.05$.

Depending on the results of separate heating/cooling experiments, Fig. 4e,f may use anomalies over the Maritime continent along X Axis rather than anomalies in the red box (this makes more sense from Response Fig. 1b anyway)

As you suggested, we re-made the scatter plot showing the relationship between domain-averaged precipitation over the Maritime Continent (MC) and corresponding spatial correlations in the mid-high latitudes derived from the CESM-LE 40-member historical simulation (Response Fig. 2). There is a marginally significant relationship between the MC precipitation trends and the spatial correlations of the circulation patterns, and the correlation coefficient across the CESM-LE 40-member is 0.3, which is smaller than 0.46 of the same scatter plot between the ITCZ drying and the spatial correlations of midlatitude Z200 patterns. Based on this new examination and above-mentioned two additional experiments, it is clear that the ITCZ drying is more closely linked with the observed wavy pattern than the MC heating. Since this additional plot doesn't add too much new information to what we already know, we decided not to include them in the revised manuscript.

Response Fig. 2. (a) The differences of precipitation (unit: mm/day/decade) trends between the members with the five largest (red markers in (b)) and five smallest (blue markers in (b)) spatial correlations in Fig. 4a of the main text. (b) The scatter plot shows the relationship between domain-averaged precipitation over the Maritime Continent (10° S-5° N, 110° E-150° E) and spatial correlations of non-zonal Z200 within the extratropics in Fig. 4a. The cross-hatched areas in (a) denote the significant differences based on the two-sample t-test, $p < 0.05$.

Fig. 1b and Resp Fig4 show that wave-like anomalies over the extratropical Pacific are displaced further north compared to the latitude of the wave pattern over the rest of the hemisphere. It reminds me the pattern in Fig. 1c in Choi and Kim, 2019. Did you think that there might be two interacting waves: one in the Pacific, triggered by anomaly over the Maritime continent, and a circumglobal wave in the NH mid-latitudes? This is something that can be suggested reading I. 288-290, but remains vague.

Further to that, reading the Caption to fig. 7, I think that a RWS in the Western Pacific by itself cannot enhance barotropic instability around the jet exit over the Northeast Atlantic. For me, it is a good argument to support the idea of two interacting waves. Otherwise, please explain better in the Caption.

Thanks for letting us rethink this possibility and improve the caption of Fig.7. It is obvious that the midlatitude circulation trends observed over the past 40 years exhibit more complex features than the typical spatial structure of the CGT pattern. It is entirely possible that other types of wave trains in the NH work together with the CGT to form the whole trend pattern. We have discussed this possibility and cited the suggested reference in a new paragraph (lines 302 to 305) as “We should note that other types of higher-latitude wave trains triggered by the tropical forcing could also interact with the CGT and contribute to the establishment of the circulation trends. Future analysis to better understand the possible interactions of these cross-latitude wave trains is needed.” We also improve the caption of Fig. 7 to emphasize that the RWS over the Western Pacific may preferentially enhance barotropic instability over the jet exit region and have a potential to generate multiple wave trains in the extratropics.

In the response letter, Resp Fig 1b shows a large VP trend centred over the Maritime continent, that is made responsible for RWS trends further north. The

problem I have with this argument is that I don't see similar RWS trends in the SH. Why is that?

The distribution and intensity of the RWS are determined not only by the divergent winds (reflecting changes of VP to some extent) but also by the absolute vorticity of the basic state. Considering the different basic states over the regions on the SH and NH sides of the Warm Pool and a sign change of planetary vorticity from the NH to the SH, we don't expect to see a mirrored distribution of RWS in these areas. However, we can still find that the most prominent RWS associated with tropical forcings occur over Eastern Australia within the core of the subtropical jet in the SH, and the RWS over this region has been a critical source of teleconnections propagating toward the West Antarctic (Ding et al. 2012)

We'd also like to mention here that we accidentally flipped the sign of the RWS in our RWS plots of the previous versions. In the revision we have corrected this error.

Response Fig. 3. Linear trend of JJA (a) Rossby wave source (RWS; shading; unit: $10^{-11} \text{s}^{-2}/\text{decade}$) over the past 40 years from ERA5. (b) is result from simulated response of JJA RWS (unit: 10^{-11}s^{-2}) to anomalous tropical heating sources added in the CESM derived from Fig. 6 (the difference of SEN and CTL). (c) is from regression between the Z200 SCI and RWS in 1800 years of CESM-LE PI simulations. (d) is from difference of RWS between the selected 23 positive and 16 negative cases in the pseudo-ensemble of CESM-LE PI runs, respectively.

Overall, I am happy with the revised manuscript, but I believe that the paper will benefit from additional experiments with separate heating anomalies.

Thanks! We have made appropriate changes to the text and hope this revision satisfies your expectations.

Minor:

Fig. 5a: ERA5 has a problem with low stratospheric temperatures in 2000-2006. For this plot I suggest using ERA5.1 if that has not been used, though it may not be particularly important for a long term trend. (<https://confluence.ecmwf.int/pages/viewpage.action?pageId=181130838>)

We recalculated Fig. 5 by replacing ERA5 by ERA5.1 over 2000-2006. As shown in Response Fig. 4 (a) and (c), the updated results are similar to the original one, but do show a slightly enhanced warming trend from the upper troposphere to lower stratosphere (above 300hpa) and a weaker warming trend in the lower troposphere

(below 800hPa). After removing the global warming signal simulated by the CESM-LE, a warm bias in CESM (above the two standard deviations of the ensemble spreads) still represents a robust feature in the tropical troposphere. This indicates that our main finding is not influenced by this data issue.

Response Fig. 4. JJA zonal mean component of air temperature (TA) trend profile from (a) ERA5+ERA5.1 (ERA5 over 2000-2006 is replaced by ERA5.1, unit: K/decade) and (b) ERA5-only, respectively. Differences of zonal mean TA trends between (c) ERA5+ERA5.1, (d) ERA5 and the ensemble mean of CESM-LE 40-member historical runs. The shaded areas indicate the ERA5+ERA5.1 (ERA5) trends lie outside the 2 standard deviations from the ensemble mean of CESM-LE 40-member historical runs. The response Fig. 4 (b) and (d) are also shown as Fig. 5 (a) and (c).

I.303: the convection is increased over the Maritime continent, please be more accurate

Thanks for pointing this out. We rewrote this sentence as "Meanwhile, the overall suppressed convection along the ITCZ favors a weakening of the equator-to-pole temperature gradient that can further enhance the waviness of the jet through increasing the zonal wavenumber of trapped stationary Rossby waves." in lines 299 to 302.

L.402, 403: I suggest omitting 'at the same level'

Thanks for the suggestion. Corrected.

I. 403: Coriolis -> planetary

Agree. Done.

I.403: is rotational component used in this paper?

The divergent component, relative vorticity (this only depends on the rotational component) and planetary vorticity are used in the calculation of the RWS, while the rotational component is not explicitly expressed in the equation. We have removed V_{ψ} in the revision.

References:

Choi, W., Kim, KY. Summertime variability of the western North Pacific subtropical high and its synoptic influences on the East Asian weather. *Sci Rep* 9, 7865 (2019). <https://doi.org/10.1038/s41598-019-44414-w>

Ding Q, Steig E J, Battisti D S, et al. Influence of the Tropics on the Southern Annular Mode [J]. *Journal of Climate*, 2012, 25(18):6330-6348.

Reviewer #2 (Remarks to the Author):

I appreciate the authors efforts to address my comments from the previous round. I am mostly satisfied that they have adequately addressed them, but I would like to comment again on one of the aspects I commented on last time along with a few minor comments listed below. For my general comment I leave it to the authors whether they wish to address this (or to the editor to decide whether they feel this is necessary) as I don't see it as being a show-stopper to publication but just something I think would be beneficial to show.

Thank you for your positive comments on our previous revision. Below are our detailed replies to each of your suggestions. We hope that we have completely addressed all raised points in this round.

General comment: I am glad the authors investigated the GOGA, TOGA and pacemaker results in the response to my review. I accept that the precipitation response in the prescribed SST experiments is not represented well enough, but I think this would be worthwhile information for readers to have, so I think it could be worth mentioning. But the main point that I'd like to make is that the authors said that they couldn't use the pacemakers to explore the impacts of the tropical Pacific SST and precip trends because they didn't have a control to compare with. But I don't think this is true. The fully coupled simulations of the CESM1 large ensemble can be used as the control to compare with. The only difference between the fully coupled CESM1 large ensemble and the pacemaker simulations is that the pacemaker simulations have the trend in the tropical Pacific SSTs imposed. It seems like if the wave train is more like the observed in the pacemaker simulations than it is in the LENS2 simulations then it would help the authors case. The pacemaker members could be added to Figure 4a for example. If they consistently show a higher correlation than the CESM-LE members then it would indicate that having the right SST trend in the eastern tropical Pacific helps push the model toward the observed trend. If they don't there's still the possibility that discrepancies in the precip trends might be messing things up. But my main point is that I don't think the lack of a control renders these simulations unusable for this purpose because the CESM large ensemble can be the control so if they help make the case for the importance of trends in the tropical eastern Pacific then perhaps it is worth adding that additional piece of evidence in.

As you suggested, by calculating the spatial correlations of non-zonal Z200 trends in ERA5 with that in the Pacemaker and CESM-LE historical runs within the NH mid-high latitudes, we find that the ensemble mean of the Pacemaker runs ($r=0.26$) shows a better similarity with observations than CESM-LE ($r=0.16$) (Response Fig. 5). This indicates that the observed SST anomalies in the tropical Pacific can help the

model simulate circulation variability in the midlatitudes to some degree, albeit with moderate improvement. We have added Response Fig. 5a, c as new Supplementary Fig. 11 and discussed its implications in the method part as “To make good use of other available CESM1 experiments conducted by NCAR Climate Variability & Change Working Group, we also examine the ensemble means of an Atmospheric Model Intercomparison Project⁶⁵ (AMIP) type Tropical Ocean Global Atmosphere⁶⁶ (TOGA) experiment and a Pacemaker run⁶², in which observed SST anomalies are nudged within the tropical Pacific (10°S–10°N, 160°W–90°W). Historical anthropogenic forcing is imposed in both these two experiments. The rainfall trends over the Western North Pacific in the TOGA run over the period (1979-2005 of historical experiments and 2006-2017 of RCP8.5 simulations) are nearly opposite to the observed pattern since the observed rainfall change results from active atmosphere-ocean coupling in the tropics rather than a sole response of the atmosphere to SST forcing as designed in AMIP runs⁶⁷. Without a correct simulation of the tropical rainfall trend pattern, it is not very informative to further examine the response in the extratropics in the TOGA run (not shown). As we expect, the Pacemaker run captures a better rainfall trend as observed over the tropical Pacific than the TOGA run and CESM-LE. The extratropical responses are also moderately improved compared with that in CESM-LE (Supplementary Fig. 11), indicating the importance of tropical convection variability in contributing to a better simulation of midlatitude circulation trends over the past 40 years.”

To better visualize how the tropical SST forcing in the Pacemaker runs contributes to the improved simulation of midlatitude circulation trends over that by CESM-LE, the differences of linear trends of SST, precipitation and Z200 between the Pacemaker runs and CESM_LE simulations over the same period are plotted in Response Fig. 6a, b, c and d. It appears that nudging observed SST anomalies over the Eastern Pacific overly cools the SST in the entire tropical Pacific, which also substantially lowers the upper tropospheric geopotential heights in the tropics. Although the rainfall trends basically show an ITCZ drying and a MC moistening, the drying trend along the ITCZ is not intact and is meshed with some positive centers around the Date Line. Another issue is that the increased rainfall in these Pacemaker runs is much broader than the observed. Strong positive rainfall anomalies over the SPCZ is generated in the Pacemaker runs, which is not seen in observations. In the extratropics, the tropical forcing due to the nudged observed SST anomalies appear to excite a wave train that is similar to observations over the North Pacific and the North Atlantic. Thus, the above-mentioned improvement of the Pacemaker runs over the CESM-LE as shown by the spatial correlation calculation is mainly due to a better simulation of the wave train from the North Pacific to North Atlantic.

Since we primarily use CESM1 in this study, we feel that it is more appropriate to examine CESM-LE2 and the heating-imposed response in CESM2 in a future analysis to explore how anomalous tropical heating generates the wave train pattern in a new model world.

Response Fig. 5. The spatial correlations between non-zonal JJA Z200 trends derived from ERA5 and (a) CESM1 Pacific Pacemaker 20-member simulations (1979-2013), and (b) CESM-LE 40-member historical simulations within the mid-high latitudes (20° N-60° N), respectively. (c) The probability density functions (PDF) of spatial correlations from (a) and (b) are calculated to indicate an improvement of the Pacemaker runs (yellow bars) in simulating the non-zonal JJA Z200 trend pattern in ERA5 over that of CESM-LE (black bars).

Response Fig. 6. Linear trends (from 1979 to 2013) of JJA (a, e) SST (unit: K/decade) (b, f) precipitation (unit: mm/day/decade) (c, g) non-zonal Z200 (unit: m/decade) and (b, h) Z200 (unit: m/decade) derived from (a, b, c, d) the differences between Pacific Pacemaker 20-member simulations and ensemble means of CESM-LE historical runs, and (e, f, g, h) from ERA5/ERSST5, respectively.

1187: In the sentence before this it's stated that some CESM-LE members capture the observed changes better than others, even though the ensemble mean fails to replicate the observed wave pattern. Now here it is then stated that this "infers the role of the aforementioned tropical forcing in contributing to the spread of the simulated recent changes in the mid-latitude circulation". I don't think this is what infers that - it's after the subsequent discussion where this can then be inferred. At this point, I think the only thing that can be inferred from the fact that some members capture the pattern better than others even though the ensemble mean doesn't totally produce it is that internal variability may play an important role.

We have rewritten this sentence to improve the readability. The new sentence is rewritten as: "This infers that internal climate variability may play an important role

in contributing to the spread of the simulated recent changes of the midlatitude circulations” in lines 185 to 186.

I100: Maybe point to Fig 1g here when discussing the precipitation changes?

Thanks for the suggestion. We have pointed to Fig 1g in the revision when we discuss this issue.

I64: "role in the change"  "role of the change"

Corrected.

I68: "nor has aquired"  "nor has it aquired"

Agree. Done.

I192: For consistency with the way the latitudes are presented, perhaps you should put degrees symbols on the longitude range.

Thanks for the suggestion. Corrected.

I246: "which resemble the CGT"  "in a manner that resembles the CGT"

Agree. Done.

Supplemental Fig 6: typo in caption "differneces"  "differences"

Corrected.

REVIEWERS' COMMENTS

Reviewer #1 (Remarks to the Author):

I am very satisfied with the responses to the comments raised in the previous round of reviews.

I think the role of the decreased meridional temperature gradient in the Pacific, which can be suggested from the observed SST trend and a corresponding shift in the ITCZ, should be more stressed in Fig. 7a. Currently, Fig. 7a makes an impression that the convection over the MC plays the major role in the wavier jet, while experiments with separate heating suggest that convection is playing only a secondary role in Z200 trends while the meridional gradient in low latitudes must be more important. The ITCZ shift is still shown in Fig. 7a, but I'd mention the gradient somewhere.

Otherwise, I am happy to recommend the paper for publication.

It's been a pleasure to review this manuscript. I am grateful to the authors for their excellent work and attention to questions raised by reviewers.

Minor comments:

L313, "other types of higher-latitude wave trains triggered by the tropical forcing": I'd remove 'by the tropical forcing' to not limit RWSs to the tropics.

Fig. 1 & 4, caption: The plots use stippling instead of cross-hatching

Reviewer #2 (Remarks to the Author):

Enhanced jet stream waviness induced by suppressed tropical Pacific convection during boreal summer, by Sun et al

I am satisfied that the authors have addressed my comments from the previous round. I only have a few remaining minor suggestions but I think the paper is now acceptable as is.

l43: "extremities"  "extremes"

l120-121: I don't really understand what is meant here. How can they "reflect uniform global ocean warming, especially in the Arctic"

l132: "those seen"  "what is seen"

l237: It seems like panel c of Figure 5 is relevant for the temperature too here?

l241: "Arctic than"  "Arctic further than"

l289: "role to increase the waviness"  "role in increasing the waviness"

l335: It is shown in the response to reviewers that ERA5.1 looks a little different when it comes to the tropical upper tropospheric warming trends. This isn't mentioned in the data and methods. Are you using ERA5.1 now for the period over which it's relevant? If so, I think that should be stated in the methods. If not, I think you should consider it since that's the more correct one I think.

l354: "ensembles"  "ensemble"

l382: "exhibits strong similarity as observed"  "exhibits strong similarity to observed"

Reviewer #1 (Remarks to the Author):

I am very satisfied with the responses to the comments raised in the previous round of reviews.

We really appreciate your positive comments on our revised version. Please find our point-to-point responses to your remaining concerns below.

I think the role of the decreased meridional temperature gradient in the Pacific, which can be suggested from the observed SST trend and a corresponding shift in the ITCZ, should be more stressed in Fig. 7a. Currently, Fig. 7a makes an impression that the convection over the MC plays the major role in the wavier jet, while experiments with separate heating suggest that convection is playing only a secondary role in Z200 trends while the meridional gradient in low latitudes must be more important. The ITCZ shift is still shown in Fig. 7a, but I'd mention the gradient somewhere.

Otherwise, I am happy to recommend the paper for publication.

It's been a pleasure to review this manuscript. I am grateful to the authors for their excellent work and attention to questions raised by reviewers.

According to your suggestion, we have revised our schematic diagram (Response Fig. 1) to emphasize the role of the decreased meridional temperature gradient induced by tropical SST forcing over the Eastern Pacific in Fig. 7a.

Response Fig. 1. A schematic diagram illustrating the mechanism: Suppressed ITCZ convection and enhanced convection over the Maritime Continent trigger the formation of the CGT-like short-waves pattern along the jet stream (and other higher-latitude wave trains) by exciting strong local RWS over the Western North Pacific and weakening the pole-to-equator temperature gradient (step a); These changes will further trigger barotropic instability around the jet exit (step b) over the Northeast Atlantic and the establishment of the CGT over Eurasia (step c). See the main text for a more detailed description of the mechanism.

Minor comments:

L313, “other types of higher-latitude wave trains triggered by the tropical forcing”:
I'd remove 'by the tropical forcing' to not limit RWSs to the tropics.

Done.

Fig. 1 & 4, caption: The plots use stippling instead of cross-hatching

Thanks for your suggestion. We have corrected the captions.

Reviewer #2 (Remarks to the Author):

Enhanced jet stream waviness induced by suppressed tropical Pacific convection during boreal summer, by Sun et al

I am satisfied that the authors have addressed my comments from the previous round. I only have a few remaining minor suggestions but I think the paper is now acceptable as is.

Thank you for agreeing to accept our paper for publication. We have accepted all your editorial suggestions.

l43: "extremities"  "extremes"

Done.

l120-121: I don't really understand what is meant here. How can they "reflect uniform global ocean warming, especially in the Arctic"

We replaced "uniform" with "prominent" as "the SST trend patterns in the CESM-LE and CMIP5/6 multi-model ensemble means reflect prominent global ocean warming..." in the revised version of the manuscript.

l132: "those seen"  "what is seen"

Corrected.

l237: It seems like panel c of Figure 5 is relevant for the temperature too here?

We agree. We add Fig. 5c at the end of the sentence to make our discussion more accurate.

l241: "Arctic than"  "Arctic further than"

Thanks for the suggestion. Done.

l289: "role to increase the waviness"  "role in increasing the waviness"

Done.

l335: It is shown in the response to reviewers that ERA5.1 looks a little different when it comes to the tropical upper tropospheric warming trends. This isn't mentioned in the data and methods. Are you using ERA5.1 now for the

period over which it's relevant? If so, I think that should be stated in the methods.

Thanks for pointing this out. According to your suggestion, we have added Supplementary Fig. 12 in the revision (the method part) to stress that our main conclusion derived from Fig. 5 is not sensitive to whether the new version of ERA5 is used. Since the main correction of ERA5.1 is to improve the representation of upper stratospheric temperature and stratospheric humidity over the 7 years (2000-2006), our main finding that primarily relies on other variables in the troposphere, such as winds and heights, holds very well regardless of which version we use. Thus, in this study, we still use ERA5 to do all calculations.

l354: "ensembles"  "ensemble"

Done.

l382: "exhibits strong similarity as observed"  "exhibits strong similarity to observed"

Corrected.